# Short- and long-term memory of moving amoeboid cells

**Peter J. M. van Haastert** [ID] *

Department of Cell Biochemistry, University of Groningen, Groningen, The Netherlands

* p.j.m.van.haastert@rug.nl

## Abstract

Amoeboid cells constantly change shape and extend protrusions. The direction of movement is not random, but is correlated with the direction of movement in the preceding minutes. The basis of this correlation is an underlying memory of direction. The presence of memory in movement is known for many decades, but its molecular mechanism is still largely unknown. This study reports in detail on the information content of directional memory, the kinetics of learning and forgetting this information, and the molecular basis for memory using *Dictyostelium* mutants. Two types of memory were characterized. A short-term memory stores for ~20 seconds the position of the last pseudopod using a local modification of the branched F-actin inducer SCAR/WAVE, which enhances one new pseudopod to be formed at the position of the previous pseudopod. A long term memory stores for ~2 minutes the activity of the last ~10 pseudopods using a cGMP-binding protein that induces myosin filaments in the rear of the cell; this inhibits pseudopods in the rear and thereby enhances pseudopods in the global front. Similar types of memory were identified in human neutrophils and mesenchymal stem cells, the protist *Dictyostelium* and the fungus *B.d. chytrid*. The synergy of short- and long-term memory explains their role in persistent movement for enhanced cell dispersal, food seeking and chemotaxis.

**Data Availability Statement:** All relevant data are within the manuscript and its Supporting Information files. The data presented are the minimal dataset that is directly deduced from the original pseudopod data.

## Introduction

The movement of amoeboid cells is mediated by actin-filled protrusions of the cell surface, pseudopodia [1]http://journals.plos.org/plosone/article?id=10.1371/journal.pone.0005253 - pone.0005253-Pollard1. Cells are very dynamic and constantly change their shape. These cells do not move in random directions (see Box 1 for nomenclature, definitions and use of terms). As early as 1953, it was shown that in the absence of external cues cells exhibit a so-called correlated random walk [2], an observation that has been reproduced for nearly all moving cells [3–7]. Correlated means that a cell is more likely to move in a direction similar to its previous direction of movement. This tendency to move in the same direction is called persistence, and the observed duration of the correlation is the persistence time. In unpolarized amoeboid cells, the persistence time may be short, less than a minute, while in polarized cells the persistence time can be several minutes long [3, 7]. The function of a correlated random walk versus a random walk has been investigated both experimentally and theoretically: It serves a strategy for

**Funding:** The author received no specific funding for this work.

**Competing interests:** NO. The authors have declared that no competing interests exist.

## Box 1. Information box: Nomenclature, definitions and use of terms

The meaning or perception of terms such as random walk, persistence and memory may differ among the sciences. To avoid confusion, I offer the definitions used here. Amoeboid cells move by making a succession of pseudopods/steps that are extended perpendicular to the cell surface. Therefore the direction of pseudopod extension is the consequence of the position on the cell surface where that pseudopod starts in combination with the local cell curvature at that position. Correlations of direction in a sequence of pseudopods/steps is therefore based on a correlation of the start position of those pseudopods/steps.

- Random walk: describes a path that consists of a succession of random steps. Here "random" needs to be specified for movement of cells with pseudopods. Successive pseudopods are extended in random directions. Since pseudopods are extended perpendicular to the cell surface, this implies that successive pseudopods start at random positions of the cell surface, where random means that all start positions have equal probability.

- Correlated random walk: Random walk where the direction of movement at one time is correlated with the direction of movement at previous times. For cells moving with pseudopods/steps perpendicular to the cell surface, correlated means that the position where a new pseudopod starts is not random but biased by the position of previous pseudopods/steps.

- A fundamental difference between random walk and correlated random walk concerns information on the past, which is absent or not used in a random walk. In strong contrast, a correlated walk must have a mechanism to collect, store and use information on the position of previous steps. In other words, a correlated random walk has memory, while a random walk has no memory. This memory can be strong (80% bias of direction) or weak (20% bias of direction), and it can be short (only information on the previous step) or long (information on many previous steps).

- Persistence is observed in experiments and is defined as the time or number of steps that the correlation of direction is detectable.

- Memory is the information collected on the position/direction of previous steps that affects the position/direction of future steps. Here memory is studied at a molecular level, aiming at identifying the molecules and mechanisms how information is collected, stored and used to bias future pseudopods.

- Persistence and memory are not identical. Persistence is an observable property, while memory is a fundamental underlying mechanism. The observed persistence depends both on the strength and the length of the memory. For instance, if the cell remembers only the previous step (short memory), but the next step is exactly in the same direction of the previous step, the cell moves in a straight line and the observed persistence is infinite. Alternatively, if the cell remembers the average of direction of many previous steps (long memory), but the next step is only slightly correlated with this average, persistence will be small.

optimal foraging and strongly improves chemotaxis [7–9]. A correlated random walk must imply that cells have a memory of direction. In this study, memory of direction is investigated from the perspective of the extending pseudopod.

The timing and direction of pseudopod formation in amoeboid cells has been described as an ordered stochastic process [3, 10]. Two components related to persistence have been recognized. First, polarized cells have a front and a rear that are semi-stable on a minute time scale; pseudopods are rarely extended from the rear but mainly from the front leading to persistence of direction. This persistence of polarized cells is observed in many different cell types, including neutrophils, fibroblasts and *Dictyostelium*. The second component is the observation that pseudopods are extending alternatingly to the right and left; this zig-zag trajectory contributes to persistence [3, 7, 11]. The combination of polarity and alternating right/left pseudopod extension is the basis for strong persistence of polarized *Dictyostelium* cells.

*Dictyostelium* cells spontaneously form multiple dynamic patches of activated Ras [12, 13]. Pseudopods frequently start at the position of the strongest Ras patch and then grow by polymerization of branched F-actin [13]. It has been proposed that pseudopods are induced by a signaling cascade consisting of activated Ras, Rac, SCAR/WAVE, Arp2/3 inducing nucleation of branched F-actin [14, 15]. In mammalian cells a similar pathway is present possibly starting with the GTP-binding protein CDC42 [16]. Memory of cell movement may be related to the probability when and where this signaling cascade is activated. Polarized cells do not extend pseudopods in the rear (70% of the cell), because myosin filaments at the side and in the rear inhibit the induction of new branched F-actin to form pseudopods [3, 17]. In mammalian cells myosin is regulated by RhoA and the Rho-kinase ROCK [18, 19] and in *Dictyostelium* by a cGMP-signaling pathway that includes two guanylyl cyclases, and the cGMP-binding protein GbpC [17, 20]. It has been shown that memory of polarity in *Dictyostelium* is related to this cGMP-signaling pathway [3, 21]. The alternating right/left formation of pseudopods in the front has been explained by the formation of a local memory [11]. Experiments show that alternating right/left formation of Ras-GTP patches is lost in cells treated with the F-actin inhibitor Lantrunculin A (latA), suggesting that F-actin itself or one of its regulatory binding proteins mediates right/left asymmetry [22]. In addition, the analysis of a large set of mutants with defects in cytoskeletal components revealed that the phosphomimetic mutant SCAR-SD [23] still extends pseudopods from the front of the polarized cell, but has lost the alternating right/left property [22]. Together, these data suggest that memory of polarity may be related to cGMP/myosin in the rear, while right/left may be related to SCAR/WAVE/F-actin in the front.

Here the molecular mechanism of memory for persistence of pseudopod formation was investigated by addressing three specific questions: 1) which information is actually stored? 2) What are the time constants of memory in terms of learning and forgetting this information? 3) Which molecules store the information?

## Results

### Two types of memory

Fig 1A shows a track of a cell with extending pseudopods. Each pseudopod is presented as a vector with direction and length (size and growth time). The information on pseudopod extension forms the basis for the current study on memory. The displacement of this cell, defined by the midpoints of the pseudopods, shows the typical properties of a correlated random walk with a correlation/persistence of about 15 pseudopods (Fig 1B). Pseudopods that are formed at the side of an existing pseudopod are called splitting pseudopod; pseudopods formed in an area of the cell without recent pseudopod activity are called de novo [3, 24]. The sequence of

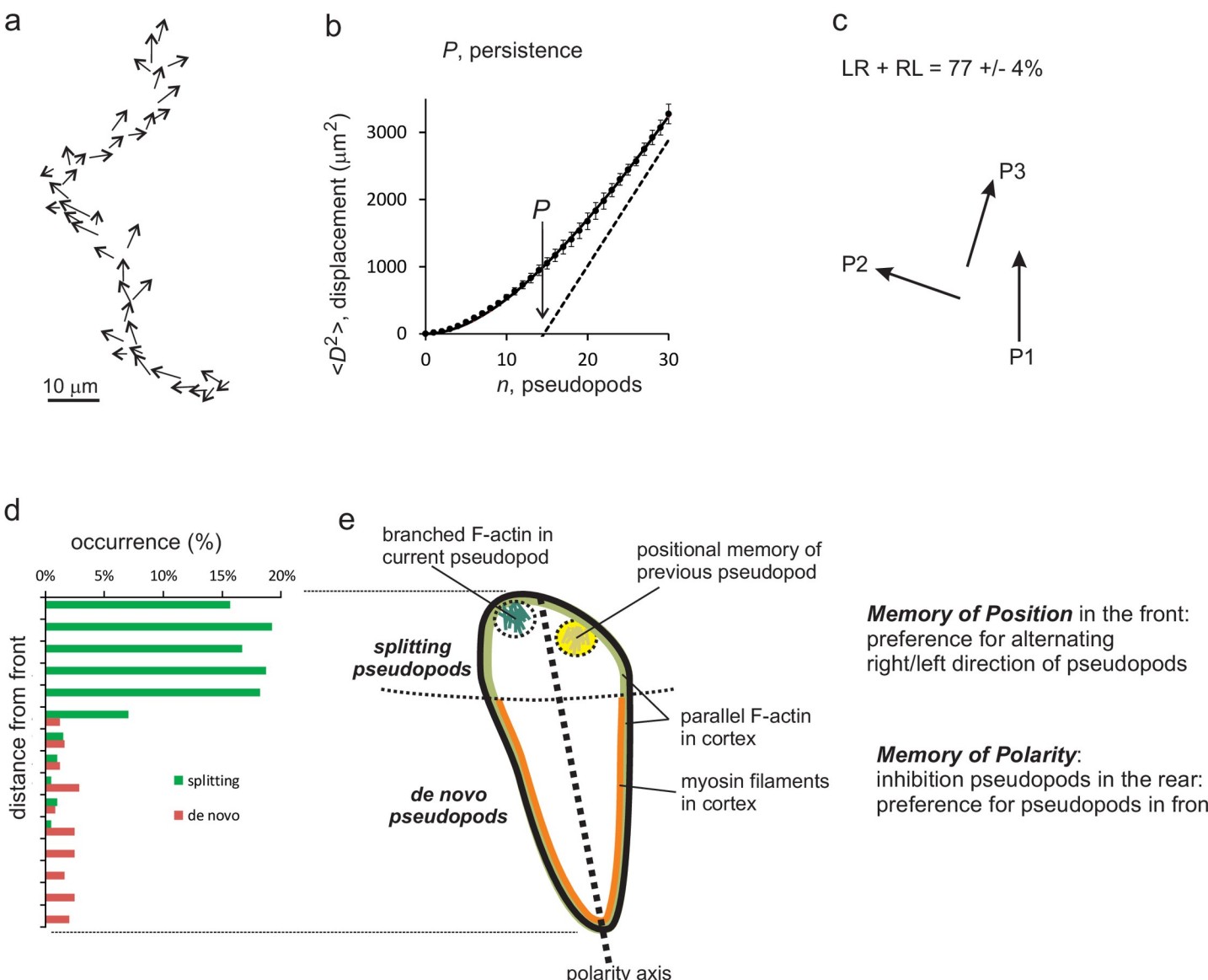

**Fig 1. *Dictyostelium* has two types of memory.** a) Path of a wild-type cell during movement in buffer for 9 minutes; the arrows indicate the pseudopods. b) The pseudopods form a trajectory of the cell. The mean square displacement ($<D^2>$) was calculated and analyzed using Eq (1), yielding a persistence of about 15 pseudopods. c) Splitting pseudopods are frequently extending alternatingly to the right and left, leading to a zig-zag trajectories with persistence. d) Cells frequently extend splitting pseudopods at the front ~30% of the cell, and rarely *de novo* pseudopods from the side or the rear of the cell. This reveals that the cell has an axis of polarity leading to persistence of direction. e) Schematic of a cell showing that the source of persistence is a combination of memory of the polarity axis enhancing pseudopods to be made in somewhere in the front, and memory of the position of previous pseudopod enhancing the next pseudopod to be made at that position in the front.

three splitting pseudopods is frequently alternating to the left and right, contributing to a persistent zig-zag trajectory (Fig 1C). Starved wild type cells are polarized with a longitudinal axis of high pseudopod inducing activity in the front and low activity in the rear of the cell. In starved wild-type cells about 85% of the pseudopods are splitting pseudopods extending from the front ~30% area of the cell, while only ~15% of all pseudopods are *de novo* pseudopods that originate from the rear ~70% area of the cell. This thus shows that pseudopod formation per area is about 10-fold more active in the front than in the rear of the cell (Fig 1D). The low activity of *de novo* pseudopod formation in the rear is due to inhibition of pseudopod formation by the acto-myosin cytoskeleton in the contractile cortex, leading to a polarized cell (Fig

1E). Together this sequence of pseudopod extensions indicates that memory has two components: First, inhibition of pseudopod formation in the rear; its associated memory is addressed as the ***memory of polarity axis***. Second, frequent alternating left-right pseudopods; as will be explained later, this memory is addressed as the ***memory of position*** of previous pseudopods.

## Input signal for right/left pseudopod splitting: Cells remember position of splitting pseudopods

Splitting pseudopods are extended alternatingly to the right and left. Thus, in a sequence of three pseudopods, the third pseudopod P3 is formed in a similar direction as the first pseudopod P1 (Fig 2A). Pseudopods are extended perpendicular to the local curvature of the cell [25]. Since a cell has a relatively smooth oval shape, this observation means that pseudopod direction and place of pseudopod start are coupled entities: two pseudopods that have started at the same place of the cell were extended in a similar direction, and *vice versa*, two pseudopods that were extended in the same direction have started from a similar position in the cell. Two very different mechanisms may explain the memory of alternating left/right pseudopod extension. In the first mechanism cells actively remember the ***direction*** of the last pseudopod and extend the next pseudopod in opposite direction. Thus, if P2 is extended at an angle $-x \pm y$ degrees to the left relative to P1, the next pseudopod P3 will be extended at $x \pm y$ degrees to the right relative to P2, and consequently P3 is extended at an angle of $0 \pm y\sqrt{2}$ relative to P1. The second mechanism is that cells actively remember the ***position*** of the previous pseudopod extension, which enhances the probability to induce future pseudopods at that position, as was suggested previously [3, 7, 11]. Thus, P3 starts at the same position as P1, and since the curvature of the cell at these positions is still nearly the same, the two pseudopods are extended in a similar direction. To discriminate between these alternative hypotheses, it is noted that for memory of direction, all pseudopods belong to one series of alternating right/left directions: P3 depends on P2, which depends on P1. Consequently, the standard deviation of the angle between P1 and P3 is larger than between P1 and P2. In contrast, the pseudopods formed with positional memory actually are decomposed into two series of pseudopods that start alternatingly at two different regions of the cell: P3 depends only on P1, and not on P2 (P2 can be far away or nearby P1/P3); therefore the standard deviation of the angle between P1 and P3 is smaller than between P1 and P2. Two very different experiments both suggest that cells memorize positional information. In the first experiment, data were collected for 322 cases of four pseudopods (P0 to P3) of which P1 to P3 are consecutive splitting pseudopods and P0 can be a splitting or de novo pseudopod. The four pseudopod vectors were rotated and transposed so that P1 always directs at zero degrees and is extended to the right relative to P0. Subsequently, the endpoint of P1, P2 and P3 are presented as dots in polar coordinates, in which all pseudopods start at the origin and the dot indicates the end of the pseudopod (see Fig 2A). The average size of all pseudopods is 6.3 ± 2.6 μm. The average angle of P2 is -52 degrees to the left relative to P1 (Fig 2B). The end points of pseudopods P3 relative to P1 is shown in Fig 2C, revealing that the average positions of P3 and P1 nearly coincide. Very important to discriminate between memory of direction or position, the variation in angle between P1 and P3 (SD = 40 degrees, Fig 2C) is much smaller than the variation in angle between P1 and P2 (SD = 62 degrees, Fig 2B) or between P2 and P3 (SD = 65 degrees, not shown). Thus, the start of P3 depends on the position of P1, not on P2. In the second experiment, the number of extending pseudopods was measured during time (Fig 3A) and analyzed by autocorrelation to find a repeating pattern of pseudopod extension. Fig 3B revels a reoccurring start of pseudopod extension with a phase of 28.7 ± 1.4 seconds (optimal fit and 95% confidence interval). Since cells extend a new pseudopod every 13.4 ± 1.6 seconds (mean and 95% CI), the data

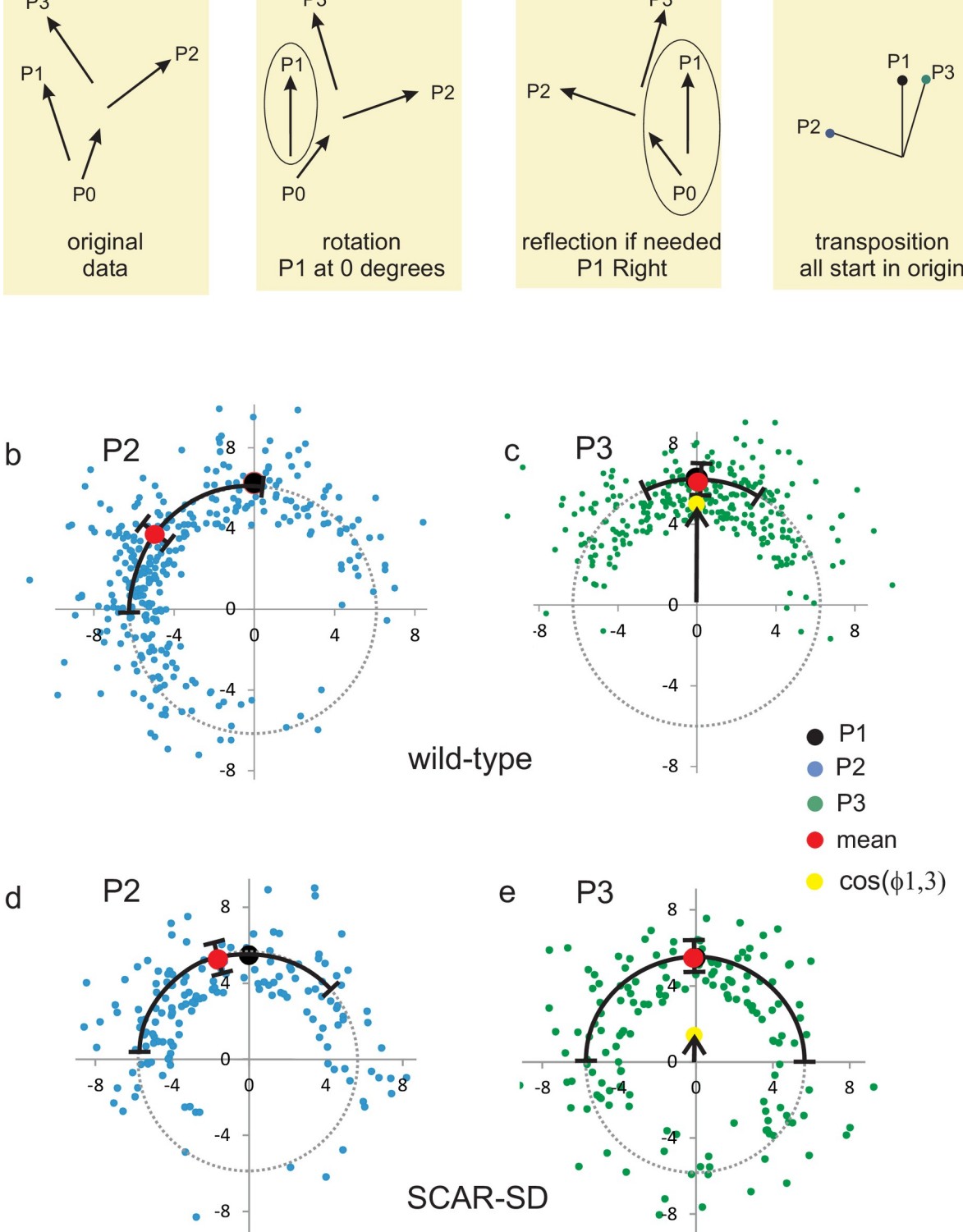

**Fig 2. Cells memorize the position of splitting pseudopods.** Analysis of four consecutive pseudopods of which P1 to P3 are three splitting pseudopods and P0 can be a splitting or a de novo pseudopod. a. Schematic for the analysis of pseudopod vectors with polar coordinates (results presented in panels b-e). All four pseudopods are rotated such that P1 directs at 0 degrees, and if needed reflected around the Y-axis such that P1 directs to the right relative to P0; finally P1 to P3 are transposed so that they start in the origin. Panels b-e: direction of P1 (black

dot; 0 degrees by definition), P2 (blue dots) and P3 (green dots). The cosine of the angle between P1 and P3 (cos($\phi$1,3), yellow dots) indicates the persistence of splitting pseudopods. b, c. Wild-type, n = 322: P2 is extended at -55 ± 62 degrees and P3 at 2 ± 40 degrees relative to P1; cos($\phi$1,3) = 0.78. d, e. mutant SCAR-SD lacking positional memory, n = 155: P2 is extended at -17 ± 67 degrees and P3 at -1 ± 90 degrees relative to P1; cos($\phi$1,3) = 0.25. The data are means and SD.

reveal that the periodic autocorrelation time is about twice the pseudopod interval. This means that the start of the first pseudopod is correlated with the start of the third pseudopod, while the start of the second pseudopod is correlated with the fourth pseudopod.

Thus, alternating left/right pseudopod splitting consists of two series with respectively the start of odd pseudopods (P1, P3, P5 etc) and the start of even pseudopods (P2, P4, P6 etc). This leads to the conclusion that splitting pseudopods remember the position of their extension.

## The kinetics of learning and forgetting the position of splitting pseudopods

In living cells, the position at the cell surface where a pseudopod has been formed is difficult to follow in time because the cell moves, but the direction of pseudopod extensions is easily monitored. Since pseudopods are always extended perpendicular to the cell surface, the angles at which pseudopods are formed provide indirectly the required information on the relative positions where pseudopods start [25]. Here the angle $\phi_{1,3}$ between pseudopods P1 and P3 is used for positional memory, and present this angle as $<\cos\phi_{1,3}>$, the average of the cosines of these angles, which is the forward movement of P3 in the direction of P1. For splitting pseudopods of polarized wild-type cells the observed $<\cos\phi_{1,3}>$ = 0.78 ± 0.05 (mean and SEM with n = 322; see yellow dot in Fig 2C). It is expected that the positional memory of P1 is formed during the extension of that pseudopod (learning), while the memory is lost at some time after pseudopod P1 has stopped (forgetting). To uncover the learning kinetics of memory, $<\cos\phi_{1,3}>$ was measured as function of the extension period of P1. Fig 4A reveals that $<\cos\phi_{1,3}>$ is small if the growth period of P1 is very short, it increased with longer growth periods and reaches a plateau of about 0.8 after about 15 seconds. The kinetic plot in Fig 4B demonstrates that learning exhibits an exponential approach to equilibrium with a half time of 3.2 ± 0.6 seconds (optimal linear fit and 95% CI). Furthermore, learning starts immediately after initiation of pseudopod P1 ($t_0$ = 0.3 ± 1.0 s; not significantly different from zero). To uncover the kinetics of memory loss, $<\cos\phi_{1,3}>$ was analyzed as function of the time interval between the termination P1 and the start of P3. Fig 4D reveals that $<\cos\phi_{1,3}>$ remains high at about 0.8 for about 25 seconds after P1 has stopped, and then declines. The kinetic plot in Fig 4E reveals that memory loss starts at 27.9 ± 2.9 seconds after P1 stopped, and then exhibits an exponential decay with a half-time of 12.8 ± 2.2 seconds (optimal fit and 95% CI).

These kinetic constants were derived from a relatively small set of the pseudopods that exhibit either brief growth of P1 (Fig 4A) or a late start of P3 (Fig 4D). Since the statistical variation in the angle $\phi_{1,3}$ is relatively large (about 40 degrees), the uncertainty on the estimates of the kinetic constants is substantial (>20%) even for this very large dataset. Estimates of the fraction of alternating L/R splitting pseudopods has a much smaller variation because it converts the analog angular signal to a digitized L or R signal. The kinetics of learning and forgetting of splitting pseudopods was also analyzed using %LR as indicator for this memory (see S1 Fig). The obtained kinetic constants for memory of %LR are very similar but with much smaller uncertainty (<10%) to those presented for $<\cos\phi_{1,3}>$: a half time of learning of 3.6 ± 0.3 seconds, and a half-time of forgetting of 14.6 ± 1.1 seconds that starts at 28.1 ± 0.6 seconds after stopping of P1 (optimal linear fit of the data and 95% CI). These kinetic constants on memory of position are consistent with the observations of Cooper et al. [11].

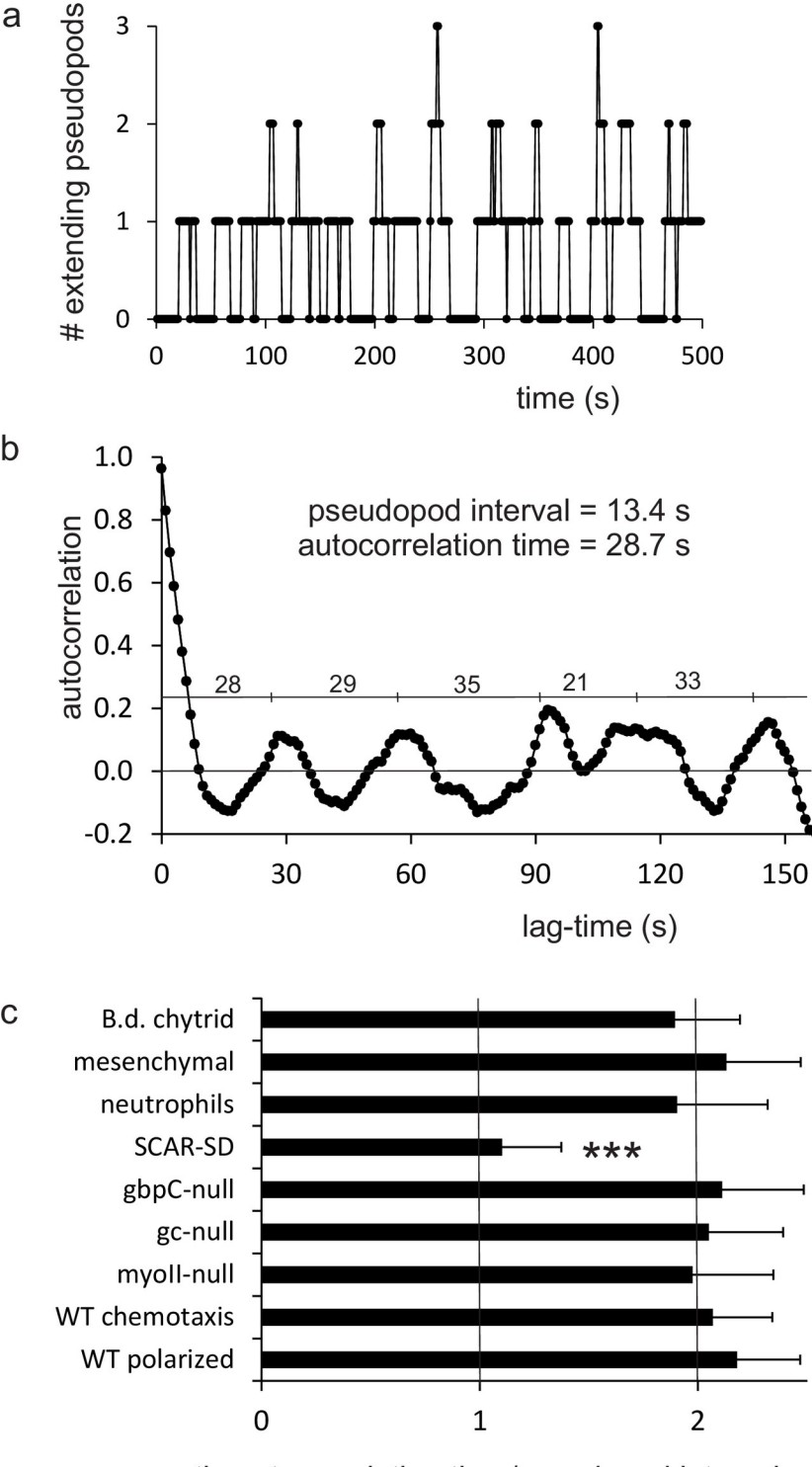

**Fig 3. Autocorrelation of the number of extending pseudopods.** a. Number of extending pseudopods of polarized wild-type cells during movement for 15,400 s; a small window of 500 s is shown. b. Autocorrelation of the number of extending pseudopods in polarized wild-type cells during 15,400 s; numbers indicate the time period between the maxima. The autocorrelation time is 28.7± 1.4 s (mean and 95% CI; n = 8). c. Ratio of autocorrelation time/pseudopod interval. The autocorrelation time of mutants and strains were determined as in panel b. The pseudopod interval was derived from [15]. The data shown are means and 95% confidence intervals. ***, the data are significantly different

from wild-type. All data, except SCAR-SD, are not significantly different from 2.0 (P>0.1). The data of ratio of SCAR-SD is not significantly different from 1.0 (P>0.1).

## Positional memory defects in mutants

The fraction of sequential splitting pseudopods that are extended alternatingly to the left and right (%LR) was measured as indicator of memory of position. Polarized cells in the absence of positional memory may have a small bias towards alternating LR, because after a pseudopod to the left the area from which the next pseudopod can emerge is larger at the right side than at the left side (see S3 Fig); random would be 50% LR, this small bias is expected to lead to 56% LR. Starved wild-type cells exhibit 77±4%LR (means and SEM, n = 28 cells; Table 1). Although vegetative unpolarized cells do not extend many splitting pseudopods, the cases with three sequential splittings have a high %LR. Also many mutant cells have a LR-bias that is indistinguishable from wild-type cells (Table 1). The only exception is the phosphomimetic mutant SCAR-SD that has a very reduced %LR, 56±5, close to the expected very low LR-bias of polarized cells without memory of position. Detailed measurements were made in this mutant for series of three splitting pseudopods (P1 to P3), of which pseudopod P1 is extended at 0 degrees and to the right relative to P0. In wild-type cells pseudopod P2 is extended 55 degrees to the

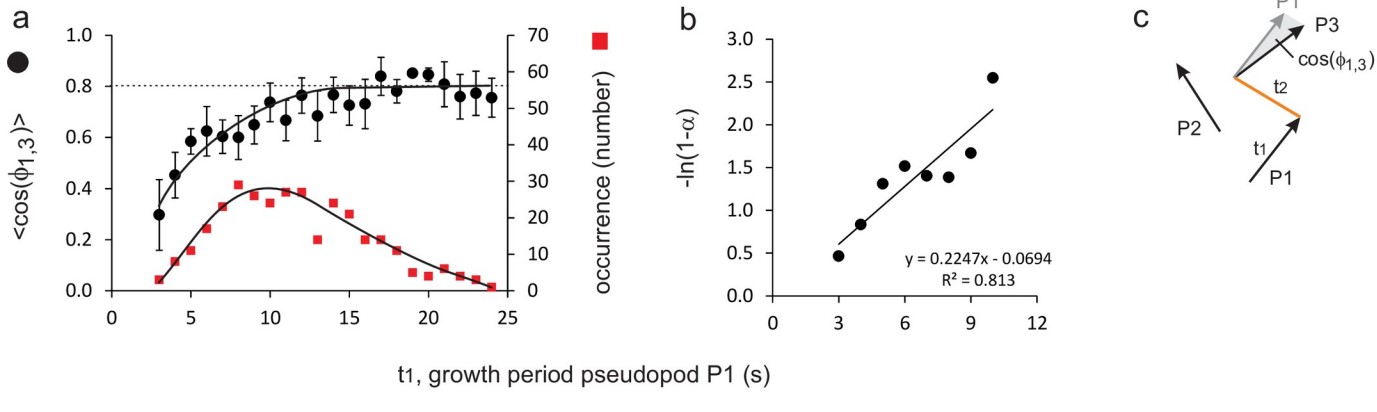

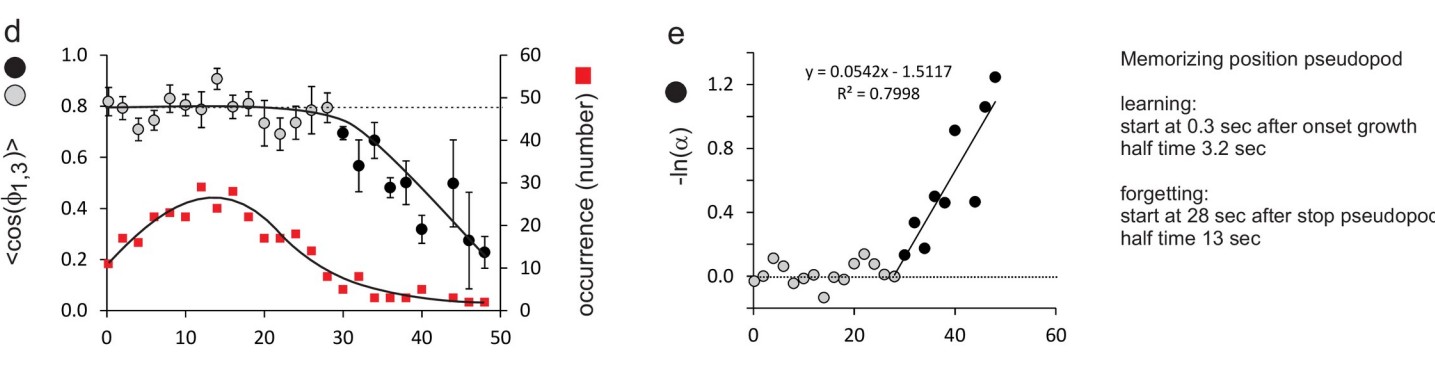

**Fig 4. Kinetics of positional memory, defined by cos($\phi$1,3).** The angle between P1 and P3 was measured for different growing times of pseudopod P1 (learning time t1 in panels a, b) and for different intervals between stop of P1 and start of P3 (forgetting time t2 in panels d, e). A small angle between pseudopods P1 and P3 (i.e. cos($\phi$1,3) close to 1.0) implies that pseudopods P1 and P3 are extended in the same direction. Panels b and e are logarithmic transformation of the data in which $\alpha = \cos(\phi_{1,3})_{t-t0} / \cos(\phi_{1,3})_{max}$. The kinetic constants for learning and forgetting were obtained by linear regression analysis; the data points represented by the black bullets in panels d and e are incorporated in the linear regression. See S1 Fig for a similar analysis using L/R bias instead of cos($\phi$1,3).

**Table 1. Memory of position and polarity axis in *Dictyostelium* mutants and other species.**

| Strain | Memory Position | | Memory Polarity-axis | | Effect of memory | |
|---|---|---|---|---|---|---|
| | alternating left-right (%LR) | | pseudopods in front (%F) | | Persistent steps (# pseudopods) | |
| ***Dictyostelium*** | | | | | | |
| Starved WT | 77±4 | | 86±6 | | 14.9±1.0 | |
| Vegetative WT | 74±2 | ns | 55±7 | * | 5.7±1.6 | * |
| Starved *sgc/gca*-null | 72±5 | ns | 41±6 | * | 9.1±1.9 | * |
| Starved *gbpC*-null | 75±4 | ns | 51±7 | * | 9.7±1.1 | * |
| Starved *myoII*-null | 73±3 | ns | 64±7 | * | 5.2±1.9 | * |
| Starved *forAEH*-null and *racE*-null | 75±4 | ns | 33±3 | * | 5.7±2.2 | * |
| Starved Rap1G12V | 76±3 | ns | 32±4 | * | nd | |
| Starved SCAR-SD | 56±5 | * | 84±5 | ns | 8.6±1.0 | * |
| Vegetative SCAR-SD | 56±3 | * | 55±3 | * | 1.4±0.4 | *, ** |
| ***Neutrophils*** | 74±5 | ns | 87±2 | ns | 13.2±1.6 | ns |
| ***Mesenchymal stem cells*** | 79±5 | ns | 86±3 | ns | 16.0±2.6 | ns |
| ***B.d. chytrid*** | 78±2 | ns | 89±4 | ns | 14.7±0.6 | ns |
| Expected if no memory | 50 to 60% | | 30 to 40% | | 1 | |
| | (see S3 Fig) | | (size of front; Fig 1E) | | | |

Data for memory of position and memory of polarity axis are the means and SEM; n = 28 cells for starved wild-type (WT), and n = 8 cells for others. Data for effect of memory are the optimal fit and 95% CI from S4 Fig and S1 Table.

ns, not significant at P>0.05

*, significantly different from starved WT at P<0.01

**, significantly different from starved SCAR-SD at P<0.01; nd, not determined because cells do not displace.

left, and pseudopod P3 is extended from a position that is very close to the position where P1 was extended (with small variation of SD = 40 degrees, see Fig 2B, 2C). As a consequence, the forward movement is very large with $<\cos\phi_{1,3}> = 0.78$. In contrast, in the mutant SCAR-SD, the extension of P2 has only a very small left bias, and P3 is extended often very far from the position where P1 was extended (large variation of SD = 92 degrees, see Fig 2D, 2E). In these polarized SCAR-SD mutant cells, pseudopods are extended in the entire front part of the cell, but not in the restricted alternating left/right areas of the front as in wild-type cells. As a consequence, the forward movement is very small with $<\cos\phi_{1,3}> = 0.2$. This indicates that SCAR-SD cells have lost positional memory of the splitting pseudopod. Finally, autocorrelation of the start of pseudopod extension was used to uncover the phase of pseudopod extension (Fig 3). For all mutants, the autocorrelation time is close to twice the pseudopod interval, demonstrating decomposition of pseudopods in two series of odd and even pseudopods, respectively (Fig 3C). Importantly, the autocorrelation time of mutant SCAR-SD is only 9.6 s close to once the pseudopod interval of 8.7 s, confirming the loss of alternating LR pseudopod extension and the loss of positional memory. This indicates that the phosphomimetic SCAR-SD disturbs positional memory, suggesting that in the signaling cascade to pseudopod extension (activated Ras → Rac → SCAR/WAVE → arp2/3 → branched F-actin), memory of position occurs at or downstream of SCAR/WAVE.

## Input signal for polarity axis: Cells remember total activity of pseudopods

The front-rear polarity axis leads to the suppression of *de novo* pseudopods at the side and in the rear of the cell [3, 24]. However, occasionally cells extend a *de novo* pseudopod in the rear

half of the cell that can form a new polarity axis (Fig 5A); what is different in cells when they extend a *de novo* pseudopod? Fig 5B reveals that just before a cell extends a *de novo* pseudopod in the rear, the pseudopod in the front is extended at a low frequency and grows unusually slow. This indicates that pseudopod formation in the front is stalled just before the extension of a *de novo* pseudopod. The experiment suggests that pseudopod activity in the front could be the input signal to generate, memorize and maintain a polarity axis. Low pseudopod activity weakens the polarity axis, thereby increasing the probability that a *de novo* pseudopod can form in the rear. In the next paragraphs this hypothesis is supported by showing that the signaling molecules that mediate polarity are formed in the front of the cell.

## Polarity defects in mutants

The fraction of the next pseudopods that start in the current front (%F) is used as indicator for the strength of the polarity axis. In the absence of polarity, cells extend many *de novo* pseudopods at the side and in the rear; the expected value %F is proportional to the relative size of the front, 30 to 40% (see legend to Table 1). For starved wild-type cells the observed value of %F is 86±6 (means and SEM, n = 28 cells). Vegetative unpolarized wild-type cells have decreased %F, indicating that suppression of *de novo* pseudopods in the rear is not present in vegetative cells. The elongated shape of starved *Dictyostelium* cells has been associated with the presence of myosin filaments at the side and rear of the cell that forms a contractile cortex with parallel F-actin. Myosin filaments formation is activated by a cGMP signaling cascade [20]. The contractile cortex is formed by multiple formins that are regulated by RacE [26]. Activated Rap1-GTP inhibits myosin filament formation [27] and possibly inhibits the entire contractile cortex [15]. Mutants defective in these pathways were investigated. Mutants lacking two guanylyl cyclases, the cGMP-binding protein GbpC or myosin II have an unpolarized shape and %F is reduced to about 50% (Table 1). Mutants lacking three formins (*forAEH*-null) or RacE extend many pseudopods from all over the cell; %F = 33%, close to the expected value for cells lacking a polarity axis. A wild-type cell overexpressing dominant active Rap1G12V show the same behavior as *forAEH*-null and *racE*-null cells with very low %F.

Fig 5B suggests that the strength of the polarity axis may depend on pseudopod activity in the front. How does the cGMP signaling pathway record pseudopod activity to form a polarity axis and how does the Rap1 pathway established polarity direction? In living polarized cells cGMP is produced predominantly by the guanylyl cyclase sGC that is localized partly in the cytoplasm and partly in F-actin rich protrusions ([28] and Fig 5C). Furthermore, in a cell lysate sGC protein is present in both the soluble and the membrane/cytoskeleton fraction, but $Mg^{2+}$-dependent sGC activity is only detected in the membrane/cytoskeleton fraction of the cell lysate [29]. This leads to the hypothesis that the observed enrichment of sGC protein in F-actin rich protrusions leads to enhanced sGC activity and elevated cGMP levels *in vivo*. To test this hypothesis cGMP levels were measured in two experiments: Localization of sGC in F-actin rich protrusions requires the N-terminal segment of the protein ([28] and Fig 5C); deletion of this N-terminus is associated with a significant decrease of cGMP levels (Fig 5D). Furthermore, addition of the F-actin inhibitor Latrunculin A (LatA) leads to the loss of sGC localization in the cortex (Fig 5C) and a drop in cGMP levels. As a control, cellular cAMP levels were measured, which are not affected by LatA (Fig 5D). These observations support the hypothesis that pseudopod activity leads to enhanced cGMP levels. It should be noted that cGMP diffuses rapidly in the cell [29], and that cGMP induces an increase of myosin filament formation in the entire cell [17]. However, cell polarity is associated with increased myosin filaments in the rear and reduced myosin filaments in the front of the cell [30], suggesting that cGMP is not sufficient to form a polarity axis. Rap1 appears the second component for

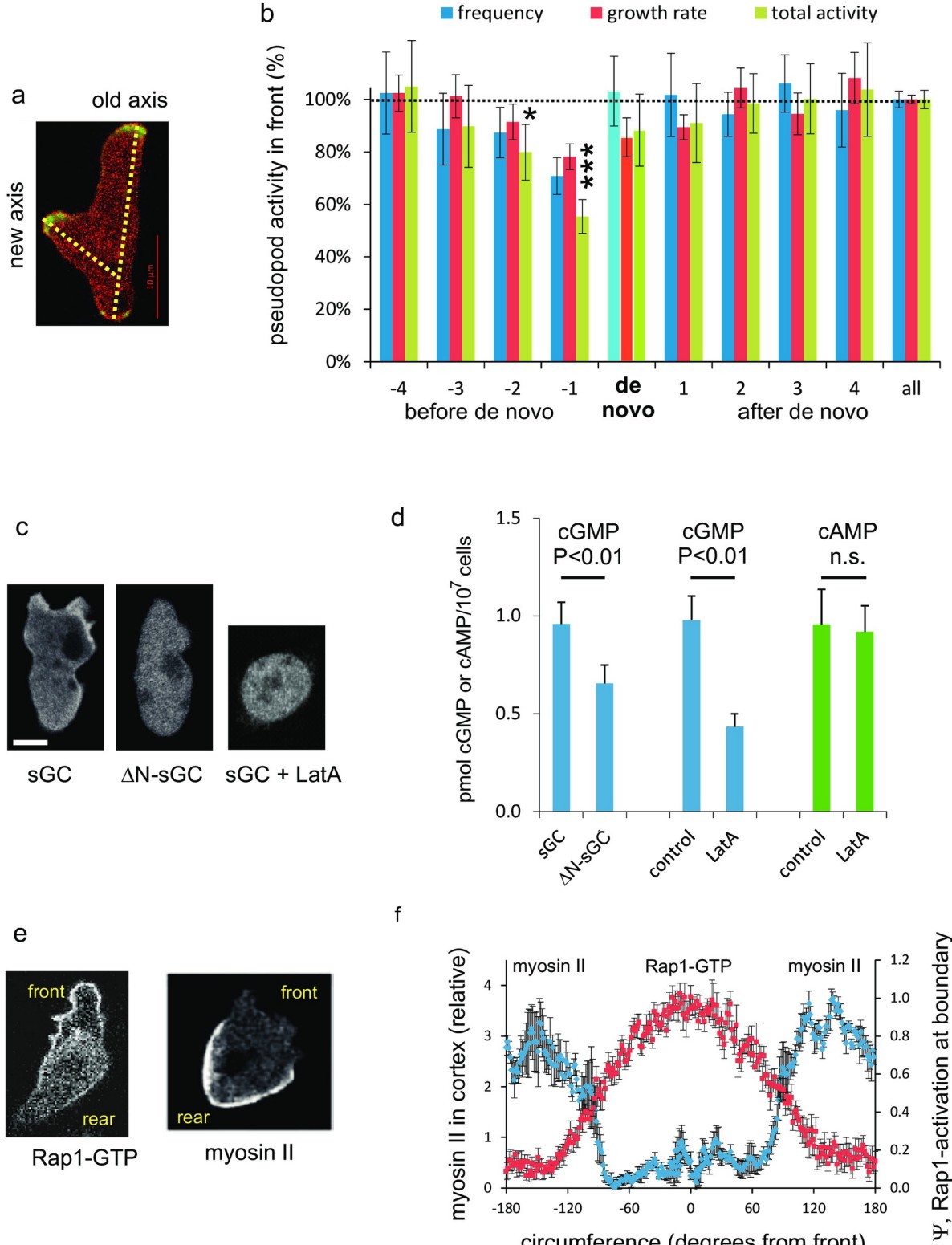

**Fig 5. Cells memorize the polarity axis by pseudopod activity and sGC activity in the front.** a. Wild-type cells expressing LimE-GFP (detecting F-actin) and myosin II-RFP. A *de novo* pseudopod can generate a new polarity axis. b. Pseudopod activity in the front before and after the extension of a *de novo* pseudopod. Measured were the pseudopod frequency (in 1/s), pseudopod growth rate (in μm/s) and their product. Data are the means and SEM of 98 *de novo* pseudopods. ***, significantly different from the data value for all pseudopods at

P<0.01; *, significant at P<0.05. c. Localization of sGC guanylyl cyclase. The left panel shows enrichment of sGC-GFP in the F-actin cortex of protrusion. This cortical localization is lost upon deletion of the N-terminus of sGC [29], or by addition of the F-actin polymerization inhibitor Latrunculin A (LatA). d. basal *in vivo* cGMP levels of *gc*-null cells expressing full length sGC or the N-terminal deletion mutant ΔN-sGC, and basal cGMP and cAMP levels of wild-type AX3 cells in the absence or presence of 10 μM LatA. Data are the means and SD of four determinations in triplicate. e. Localization of Rap1-GFP (using Ral-GDS-GFP and cytosolic-RFP [22]) and myosin II (with myosin II-GFP in cells under agar [33]). f. Fluorescent intensity of Ral-GDS-GFP minus cytosolic-RFP (Ψ) representing Rap1-GTP levels at the boundary of the cell (mean and SEM of 10 cells). Fluorescent intensity of myosin II-GFP in the cortex relative to the intensity in the cytosol (mean and SEM of 5 cells).

polarity: Active Rap1-GTP leads to depolymerization of myosin filaments [27] and is localized in the front 50% of the cell in a much broader area than F-actin, Ras-GTP and Rac-GTP that are localized in the utmost 15% front of the cell [22]. Importantly, myosin filaments are localized in the rear 50% of the cell showing the mirror image of Rap1-GTP localization (Fig 5E and 5F) [31–33].

To further investigate the role of these signaling molecules for the polarity axis, the levels of F-actin, myosin, Ras-GTP and Rap1-GTP were measured with sensitive dyes in polarized wild-type cells that either form a new polarity axis, or retract an existing polarity axis. When a cell extends a *de novo* pseudopod in the contractile cortex at the side of the cell, several events occur before pseudopod extension starts (Fig 6A). The first event at the place of the future pseudopod is the activation of Ras (at –6s relative to the start of the pseudopod), followed by the activation of Rap1 (at -4s) and the disappearance of myosin (at -3s). F-actin accumulates at –1.6s relative to the start of the *de novo* pseudopod. This sequence of events is compatible with a signaling pathway of spontaneous Ras-GTP patches that first activates Rap1 leading to myosin disassembly to locally weaken the contractile cortex and later activates the formation of branched F-actin to start the *de novo* pseudopod. Fig 6C shows the localization of F-actin and myosin in a cell with two extending pseudopods (upper panel), of which one pseudopod stops (middle panels) and is retracted (lower panels). In all ten pseudopods, of ten comparable cells, that later are retracted the first event is a decline of F-actin followed about 3s later by the arrest of pseudopod extension (Fig 6B). Then the pseudopod remains stationary for very different times (from 4 to 20 seconds for the ten pseudopods analyzed), followed by active retraction. The first event in this retraction phase is the inactivation of Ras-GTP (at –5.5s before retraction) followed by the inactivation of Rap-GTP (at –4s) and the accumulation of myosin filaments (at –2s). This sequence of events is compatible with the hypothesis that Ras-GTP and Rap1-GTP must disappear to allow myosin filaments to be formed and retract the old pseudopod.

In summary, polarity in *Dictyostelium* is regulated by two signaling pathways that both start in the front of the cell and depend on pseudopod activity. In the first pathway, Ras-GTP in the front of the cell induces F-actin filled protrusions; sGC activity is enhanced in these protrusions and possibly further stimulated by Ras-GTP [34]. Produced cGMP rapidly diffuses in the cell, activates GbpC leading to an increase of myosin filament in the entire cell. The second pathway also starts with Ras-GTP in the front and mediates the activation of Rap1 in the front half of the cell. Rap-GTP leads to the local disassembly of myosin filaments and instability of the contractile cortex in the front half of the cell. The cGMP signaling pathway thus provides information to establish the polarity, but it does not determine the direction of polarity. The direction of the polarity axis is provided by Rap1-GTP.

## The kinetics of learning and forgetting of the polarity axis

The polarity axis leads to the suppression of new pseudopods in the rear of the cell. When a cell extends a *de novo* pseudopod in the rear, the cell gets two fronts, which ultimately is not

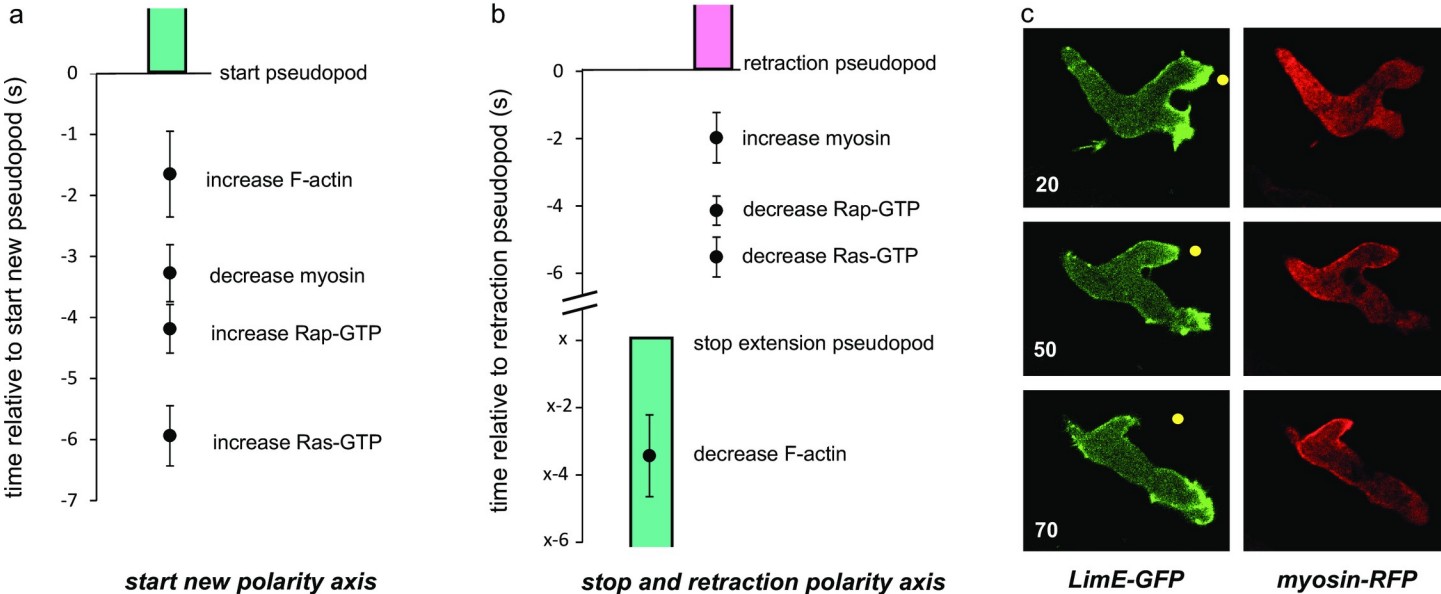

**Fig 6. Kinetics of signaling molecules in pseudopods of a new or a retracting polarity axis.** Wild type AX3 cells expressing different markers were analyzed. a. Cells begin to extend a *de novo* pseudopod at the side of the cell forming a new polarity axis. Time t = 0 is the start of the pseudopod. Indicated are the times at which the sensors for the indicated signaling molecules begin to increase or decrease at the place where the pseudopod later begins. Data are means and SEM of 8 determinations. b. Cells with two extending pseudopods retract one pseudopod. The extension of one pseudopod stops at time t = x and retraction begins at t = 0s. The value of x, is very variable, between 4 and 20 s, but in all cases F-actin declines about 3 s before the pseudopod stops. Data are means and SEM of 10 determinations. c. Images of a cell expressing LimE-GFP (detecting F-actin) and myosin-RFP that retracts the upper pseudopod. The yellow dot is at identical positions in the three images. The upper pseudopod begins retraction at 50s.

stable. Therefore, three situations have been detected. First, the new pseudopod/front has no or very little additional pseudopod activity and is soon retracted. Second, the new pseudopod/front has strong pseudopod activity and the old front is retracted. And third, both old and new front have pseudopod extension activity by which two competing fronts appear of which one front finally has to retract. Fig 7A shows a time series with pseudopod activity in both the old and new front, and finally retraction of the old front. Fig 7B shows an example where only the new pseudopod exhibits pseudopod activity, while the old pseudopod remains silent and is retracted. Pseudopod activity in combination with the time of retraction of the Old or New front was used to determine the kinetics of memory formation of the New axis and memory loss of the Old axis.

A simple model for polarity axis was constructed, in which the strength of the polarity axis increases with pseudopod activity in its front, and decreases in the absence of pseudopod activity (see methods). Based on this model, it is to be expected that shortly after formation of the new *de novo* pseudopod, the New axis is weak and the Old axis is still strong; in time the Old axis may become weaker and the New axis stronger, depending on the pseudopod activity in their fronts. Thus, it is expected that in cases of early retraction mainly New fronts are retracted, while retractions of Old fronts prevail in very late retractions. This is demonstrated in Fig 7C showing all 98 observations of formation of a New axis. Most retractions occur between 20 and 60 seconds after formation of a New front; early retractions are almost always the New front, while the Old front increasingly is retracted at later times. New and Old fronts have equal probability to be retracted after about 48 seconds, suggesting that the loss of strength of the Old front matches the gain of the New front at this time moment.

These 98 cases are very diverse in the pseudopod activity of the Old and New front. For determining exact kinetics, only those 28 cases were used in which the Old front does not

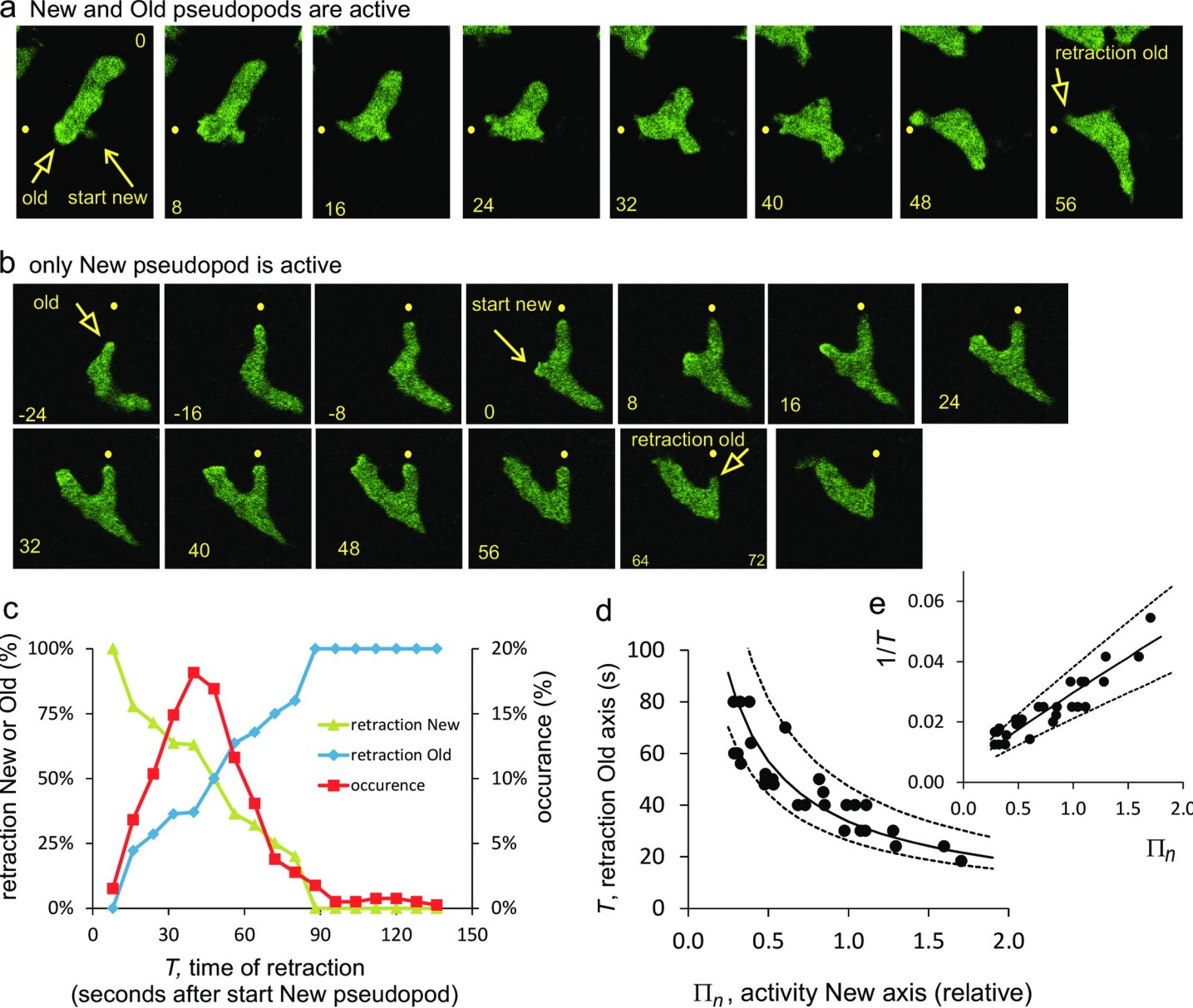

**Fig 7. Kinetics of memory of polarity axis.** Wild-type AX3 cells expressing Raf-RBD-GFP are moving in buffer. At t = 0 seconds the cells extend a *de novo* pseudopod, generating a NEW front. Pseudopod activity (seize and frequency) was measured in the New and Old front, as well as the time of retraction (T) of the pseudopod in the Old or the New front. Panels a show an example in which both the Old and New front exhibit pseudopod activity. Panels b show an example in which only the New front exhibits pseudopod activity. c. All 98 cases of a *de novo* pseudopod were analyzed for the retraction of either the Old or the New front. d. Detailed analysis of the subset of 28 cases in which only the New front exhibits pseudopod activity and the pseudopod in the Old front is retracted at time T. The experimental data were fitted to Eq (7) yielding a half-time of learning (25.7 ± 4.6 s) and forgetting (88 ± 43) of the polarity axis (optimal fit and 95% CI). The full line is the optimal fit, the dashed lines give the 95% confidence levels.

exhibit any pseudopod activity after the New front was made. In these cases the memory of the Old front is no longer enforced and therefore declines exponentially, whereas the memory of the New front increases with time according to its observed pseudopod activity. Thereby, the model describing polarity reduces to a model with solvable kinetic equation (see methods). Pseudopod frequency and pseudopod growth were measured in the New front (designated as $\Pi_n$) and the retraction time T of the Old front. The experiments presented in Fig 7D and 7E

reveal an inverse relationship between $T$ and $\Pi_n$: the Old axis is retracted faster when the New axis has more pseudopod activity. According to the model for the formation of the polarity axis, the relationship between retraction time $T$ and pseudopod activity $\Pi_n$ is described by (see methods)

$$\frac{1}{T} = (aP\Pi_n + b)/\emptyset, \quad \text{where } \emptyset = -\ln\left(1 - \frac{0.9(a\Pi_n + b)}{\Pi_n(a+b)} e^{-bT}\right) \tag{7}$$

where $a$ and $b$ are the kinetic constants of learning and forgetting, respectively. The observed inverse relationship between $T$ and $\Pi_n$ suggests that $\phi$ does not strongly depend on $\Pi_n$ and $T$. Calculations using observed values for $T$ and $\Pi_n$ reveal that $\phi$ depends predominantly on the kinetic parameters $a$ and $b$, and only weakly on $\Pi_n$ and $T$. To find estimates for the kinetic constants $a$ and $b$, $1/T$ was calculated with Eq (7) using the observed values of $\Pi_n$ and different values of the kinetic constants $a$ and $b$. The optimal fit between calculated and observed value of $1/T$ yielded the optimal values of a and b. The accuracy of the fit was estimated using bootstrap analysis of the data, yielding the 95% CI for a and b (see methods). This analysis provide estimates of the mean half-time for formation of the New polarity axis (25.7 ± 4.6 seconds; range 22 to 31 seconds, optimal fit and 95% confidence level) and the half-time of forgetting of the Old polarity axis (88 ± 43 seconds; range 63 to 148 seconds). Cells extend on average one pseudopod per 15 seconds, which means that half-maximal learning of a polarity axis requires the extension of 1 to 2 pseudopods, while half-maximal forgetting of the old axis requires the absence of 4 to 10 pseudopods. Compared to the learning and forgetting kinetics of the *memory of position*, the kinetics of *memory of the polarity axis* is about 4 to 8-fold slower.

## Adjustment of the polarity axis by pseudopods in the front

In general, the polarity axis and the memory of position are congruent: pseudopod P3 starts at a similar position as P1, ignores the direction of P2 (Fig 2C), and therefore the polarity axis maintains its direction. However, sometimes P2 is extended at a very large angle relative to P1. Does this wide angle deflection change the direction of the polarity axis and the position of P3? Fig 8A reveals that relationship between the angles $\phi_{1,2}$ between P1 and P2, and the angles $\phi_{2,3}$ between P2 and P3 is well fitted by a linear regression ($\phi_{2,3}$ = -1.0027 $\phi_{1,2}$); however, this relation is statistically better fitted by a polynomial ($\phi_{2,3} = 1.64^*10^{-5} \phi_{1,2}{}^3 - 1.0027 \phi_{1,2}$). Using the Akaike Information Criterion (AIC) the polynomial model describes the experimental data 90-times better than the linear model (see legend Fig 8A). This suggests that in extreme cases where P2 is extended very far away from P1 (at a very large angle), P3 is not extended at the position of P1, but slightly biased towards P2; this bias is indicated by the green area in Fig 8A. Up to an angle of $\phi_{1,2}$ of 80 degrees, P3 still starts in the direction of P1, but when P2 is extended at larger angles a small deviation occurs of 11 degrees at $\phi_{1,2}$ = 90 degrees, 28 degrees at $\phi_{1,2}$ = 120 degrees and 45 degrees at $\phi_{1,2}$ = 150 degrees. It should be mentioned that such extreme positions of P2 are rare; only 12% of the pseudopods in the front have $\phi_{1,2}$ >90 degrees.

The localization of Rap1-GTP was monitored with Ral-GDS-GFP/cyt-RFP in cells that extend a new pseudopod with large deviation from previous pseudopod. In the cell images (Fig 8C), the polarity axis is given by the line connecting the uropod and the maximal levels of Rap1-GTP. In this example the cell extends a new pseudopod at an angle of 130 degrees to the left. The level of Rap1-GTP increases in the new pseudopod, but since the level of Rap1-GTP at the right side of the cell does not change, the maximal level of Rap1-GTP changes by only -22 degrees to the left. This analysis was done for many cell (Fig 8D), showing that new pseudopods at an angle $\phi_{1,2}$ <80 degrees hardly change the direction of Rap1-GTP localization,

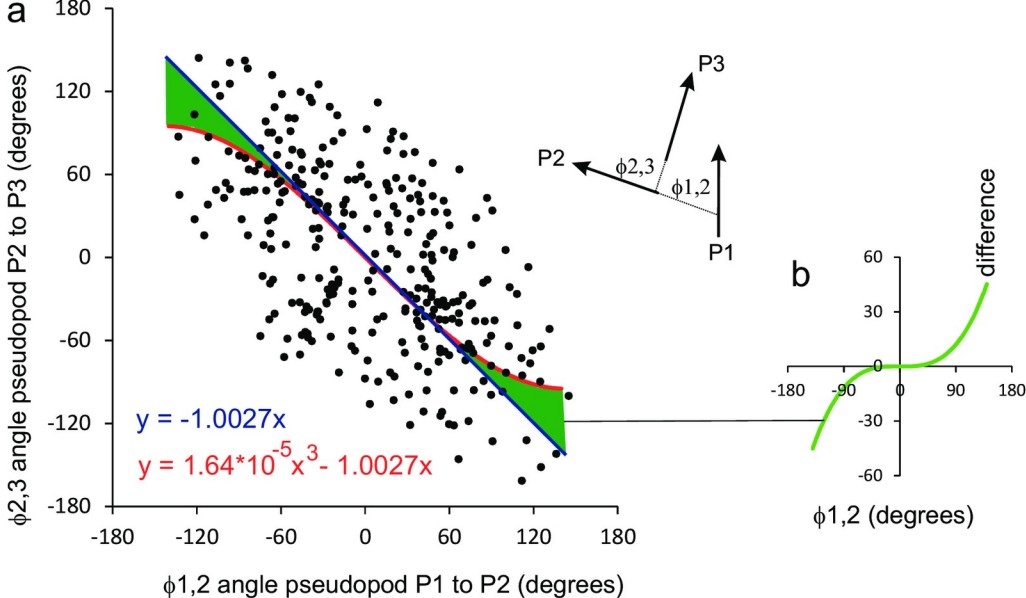

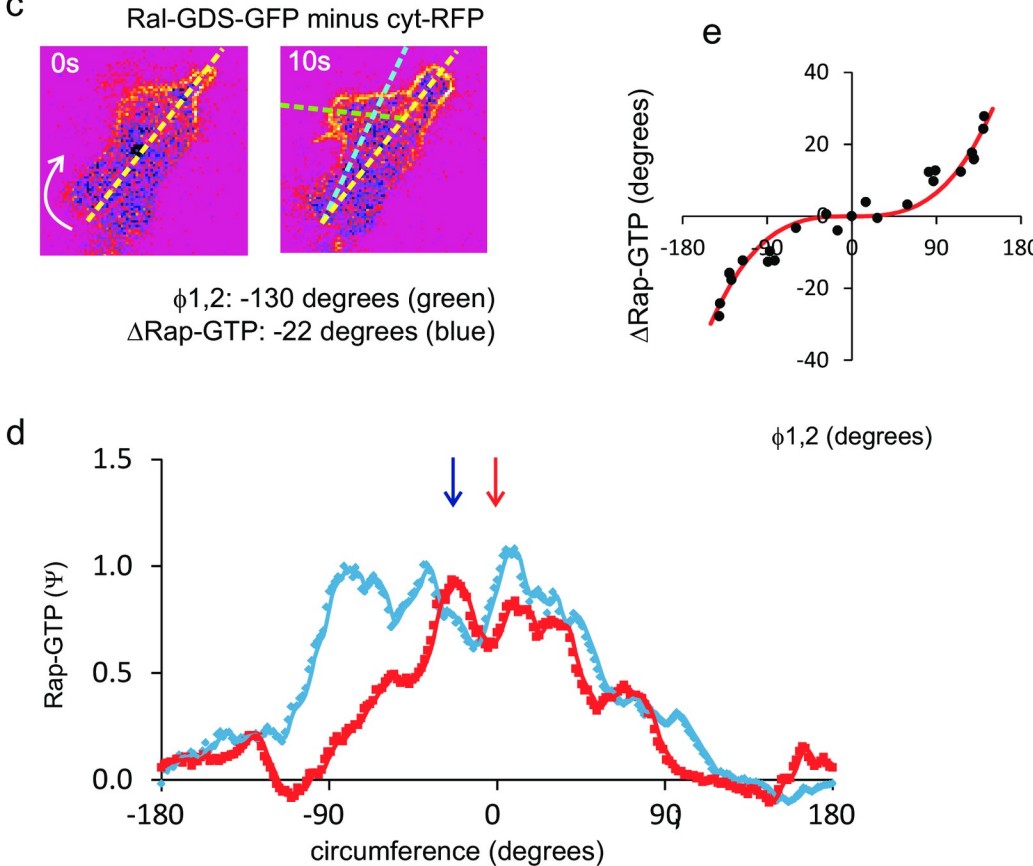

**Fig 8. Maintenance and adjustment of the polarity axis by pseudopods in the front.** a. angles between three pseudopods; a perfect inverse relationship implies that P1 and P3 are extended in a similar direction, irrespective of the direction of P2. The data were analyzed by linear and polynomial regression analysis, yielding y = -1.0027x, RSS = $4.07*10^5$ and $AIC_{c1}$ = 4841 for linear regression with one parameter and y = $1.64*10^{-5}*x^3 - 1.0027x$, RSS = $3.91*10^5$ and $AIC_{c2}$ =

4832 for the polynomial fit with two parameters. The polynomial fit explains the data significantly better than the linear fit (P = 0.01). The quantity $\exp((AICc_1 - AICc_2)/2) = 90$ indicates that the model with two parameters is 90 times as probable as the model with one parameter. The green area indicate the difference between the linear and polynomial fit, which reveals that P3 is not extended precisely in the direction of P1 if P2 is extended at a very large angle relative to P1; this difference ($y = 1.64^*10^{-5*}x^3$) is indicated in panel c. Localization of active Rap1-GTP in a cell that extends a new pseudopod from the front at a large angle of 130 degrees to the left. The fluorescence intensity at the boundary is shown in panel d, showing that the midpoint of Rap1 activation changes by 22 degrees to the left. Panel e shows the change of midpoint of Rap activation as function of the change of pseudopod direction for 12 pseudopods as determined in panels c, d. the data points are mirror imaged; the red line shows the optimal polynomial fit with $y = 0.89^*10^{-5*}x^3$. The results reveal that Rap1 is activated in the front half of the cell, and that this localization changes hardly thereby maintaining the direction of the polarity axis.

whereas Rap1-GTP localization changes by about 30 degrees at $\phi_{1,2} = 140$ degrees. The results reveal that the localization of Rap1-GTP in a broad area at the front half of the cell not only leads to the depletion of myosin filament in the front half of the cell, but that this also preserves the polarity when new pseudopods are made somewhere in the front half of the cell.

These experimental observations on the extension of pseudopods at large angles with the associated change of the polarity axis were used to predict how fast the polarity axis of a cell will turn if the cell initially moves alternatingly R/L at 0 degrees with the right pseudopod at 27 degrees and then all Right pseudopods are extending at 90 degrees. The cell will slowly turn in the new direction with a predicted half-time of 109 seconds (S2C Fig). Turning was studied experimentally using the chemoattractant cAMP. Polarized cells moving in buffer with strong persistence and a strong polarity axis were exposed to a shallow gradient of the chemoattractant cAMP that is oriented opposite to the direction of cell movement (S2D Fig). The cell does not extend a pseudopod from the back towards cAMP, but remains polarized and slowly turns at the front. Thus, the external cAMP gradient modifies the direction of the polarity axis. The time required to turn and move in the direction of the cAMP gradient is 130 ± 25 seconds (mean and SD, n = 6 cells). These data are in close agreement with those reported by Chen et al. who observed a 160 s turning time for polarized *Dictyostelium* cells [35], and Zigmond et al. who reported a 2 to 3 minute turning times for neutrophils in chemoattractant gradients [36].

Taken together, polarized cells have a memory of the polarity axis that can change direction when exposed to a new internal gradient by a *de novo* pseudopod (Fig 5), by an extreme splitting pseudopod (Fig 8), or by an external chemoattractant gradient (S2 Fig); in all cases, this change of direction of the polarity axis has a half-time of 100 to 130 seconds.

## Role of memories for persistence of cell movement

In the absence of memory each pseudopod is extended in a random direction, and therefore the cell moves in a persistent direction for only one pseudopod, which is ~10 to 30 seconds. With memory, cells move with persistence of several pseudopods. Determining the persistence with the mean square displacement (MSD) method (Fig 1A) is facilitated by having long trajectories. Therefore, long sequences of pseudopods were obtained by concatenating the pseudopods from multiple cells (see methods). The MSD of polarized cells (S4 Fig) shows the typical transition of a persistent ballistic mode of movement to a random diffusive movement; the step size is 3.1 ± 0.1 μm and the persistence is 14.9 ± 1.0 pseudopods/steps (Table 1 and S1 Table). Mutants with defective memory of the polarity axis (*gc*-null, *gbpC*-null, *myoII*-null, *for-AEH*-null, *racE*-null) have a reduced persistence of 5 to 9 steps; a similar reduced persistence is found for the starved polarized SCAR-SD mutant missing the positional memory of the splitting pseudopod. In the vegetative SCAR-SD mutant, with defects in both position and polarity memories, the persistence is only 1.4 steps, very close to the expected persistence of 1 step for

cells devoid of any memory. In summary, the suppression of *de novo* pseudopods in the rear of polarized cells leads to an increase of persistence of cell movement, because the majority of protrusions are formed in a relatively small and stable front. This persistence by the polarity axis is enhanced by the alternatingly to the right and left extension of the splitting pseudopod, because over longer times a relatively straight zig-zag trajectory is formed.

## Memories of other organisms

Recently the kinetics of pseudopod extension was investigated in *Dictyostelium*, neutrophils, mesenchymal stem cells and the fungus *Bd. chytrid* [15]. Here these pseudopod data were used to investigate the potential presence of memory; tracks with extending pseudopods are presented in S5 Fig. The fraction of pseudopods in the front 30% area of the cell is used in these cells as indicator for memory of polarity. The results presented in Table 1 reveal that all four cell lines have similar values for %F, indicating that they have a similar memory of polarity. For memory of position the fraction of alternating R/L bias was used as primary indicator. The results reveal that all four cells have similar %LR. Subsequently the number of extending pseudopods was analyzed by autocorrelation (Fig 3C), revealing that the autocorrelation time is approximately twice the pseudopod interval. Finally, trajectories of pseudopods were analyzed using MSD (S4 Fig), revealing a persistence of about 15 steps, similar to persistence of polarized *Dictyostelium* cells. Thus, although these four cells have very different pseudopod intervals [15] (see S1 Table), the pseudopods in the front have positional memory leading to correlations of the start-position of odd pseudopods and of even pseudopods, very similar as was analyzed in detail for *Dictyostelium*.

## Discussion

Amoeboid cells move with persistence: their current direction of movement is correlated with the direction in the past. This implies that cells somehow are able to store information on their direction of movement and use this information to bias future movement: cells have directional memory. The present study reveals that cells use positional information to bias direction. Cells move by extending pseudopods perpendicular to the surface curvature [25]. Therefore the direction of movement is determined by the position on the cell surface where a pseudopod is induced: a pseudopod induced in the front leads to further forward movement, whereas a pseudopod extended from the side or rear leads to sideway or reversal of movement, respectively. Therefore the molecular basis for directional memory is hidden in the memory of the position on the cell surface where pseudopods have been extended in the past. The present study has revealed that two very different mechanisms of memory cooperate to provide persistence: a long-term memory of polarity axis and a short-term memory of pseudopod position (Fig 9). Importantly, evidence was obtained that these two mechanisms of memory are present in four organisms, the fast moving chemotactic *Dictyostelium* and neutrophil cells, the slow moving mesenchymal stem cells, and the fungus *B.d chytrid*.

### Long-term memory of polarity axis

Cells are polarized with a front to rear axis of pseudopod inducing activity [37]. Parallel F-actin filaments in conjunction with myosin filaments in the cortex at the side and rear of the cell suppress the extension of pseudopods in the anterior ~70% of the cell. The majority of the pseudopods are extended from the front ~30% part of the cell. Experiments with starved mutant SCAR-SD, which possesses only this long-term memory, indicate that there is no further preference for the position of pseudopod extension within this ~30% front area (Fig 2D). Thus, the long-term memory is not precise but global within the front. The molecular

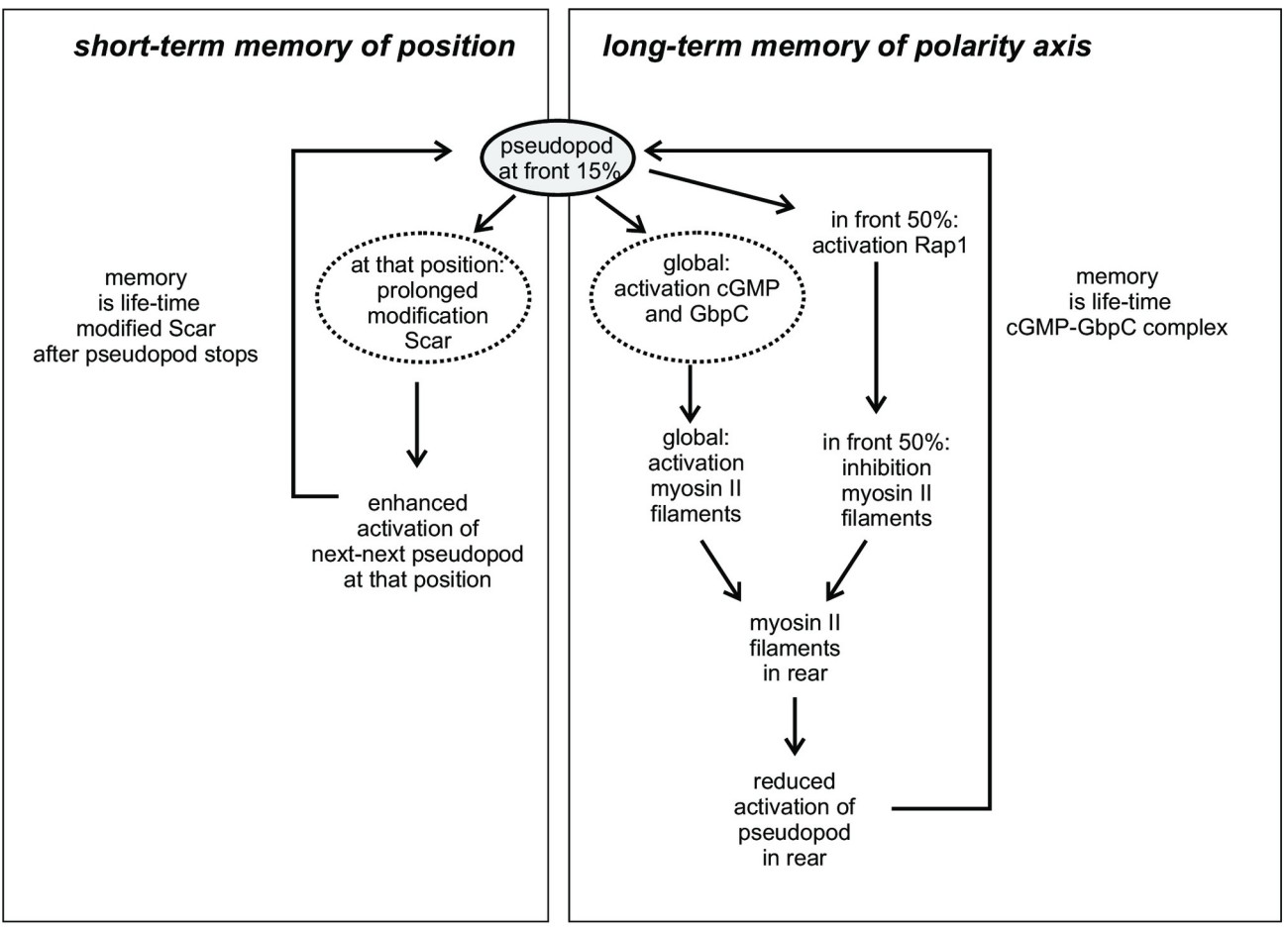

| Property | Memory Position | Memory Polarity axis |
|---|---|---|
| Input for memory | Pseudopod position (spatial) | Pseudopod activity (temporal) |
| Learning time (s) | 3.4 s | 25.7 s |
| Forgetting time (s) | 28 s refractory, then 13 s | 88 s |
| Result of memory | Activation start next-next pseudopod at same position | Inhibition pseudopod in rear half of the cell |
| Molecular mechanism | SCAR-dephosphorylation | Binding cGMP to GbpC |
| | F-actin | Myosin II |
| Accuracy or precision | Precise (6% of cell surface) | Global (front 30%) |
| Integration of information | No, one pseudopod | Yes, ~6 pseudopods |

**Fig 9. Schematic and properties of short-term memory of position and long-term memory of polarity axis.** The extending pseudopod provides signals to two memory systems. Left part: at the position of the extending pseudopod SCAR becomes modified, which enhances the probability to start the next-next pseudopod at that local position. Right part: the extending pseudopod enhances guanylyl cyclase activity generating fast diffusing cGMP in the entire cell, which binds to GbpC and induces myosin filaments in the entire cell. The extending pseudopod also induces Rap1 activation that inhibits

myosin filaments in the front of the cell. Since cGMP-binding to and dissociation from GbpC is slow, GbpC integrates information from ~10 successive pseudopods thereby generating a long-term memory of cell polarity. The Table summarizes the properties of memory of position and memory of polarity.

mechanism of cell polarization in *Dictyostelium* has been characterized [20, 21, 27, 38, 39] and is fully consistent with the properties of the long-term memory (Fig 9, right panel). Pseudopods in the rear are suppressed by acto-myosin filaments in the rear of the cell. Myosin filaments are induced in *Dictyostelium* by a cGMP-signaling pathway. cGMP produced by the guanylyl cyclases GCA and sGC activate the cGMP-binding protein GbpC, that induces myosin filaments in the cortex [20]. Since cGMP diffuses rapidly, myosin filaments are formed in the entire cortex. Ras/Rap is activated in extending pseudopods, and activated Rap induces local depolymerization of myosin filaments [22, 27] The input signal for the long-term memory is probably total pseudopod activity. The regulation of the guanylyl cyclase sGC has been well characterized. The protein is cytosolic and enriched in the cytoskeleton where it has strongly enhanced $Mg^{2+}$-dependent activity [28, 29]. We observed that the F-actin inhibitor LatA leads to a 60% reduction of cGMP levels, suggesting that the detection of total pseudopod activity for memory occurs through cytoskeleton-associated sGC activity (Fig 5D). Therefore it is proposed that extending pseudopods in the front provide two signals that determine the polarity axis: enhanced cGMP formation that induces myosin filaments in the entire cell and Rap1-GTP that inhibits myosin filaments in the front. The cGMP signaling pathway provides the signal that the cell has a polarity axis (but does not supports the direction of the axis); as long as new pseudopods are made and cGMP levels are elevated, the polarity axis remains strong. However, when pseudopod activity in the front stalls, the polarity axis weakens and a pseudopod at the side of the cell may appear with a new polarity axis. Rap1-GTP is present in a broad front encompassing nearly the half of the cell, and provides the direction of the polarity axis. The localization of Rap1-GTP changes hardly when a new pseudopod is made somewhere in this front, thereby conserving the direction of the polarity axis. Interestingly, even when a new pseudopod is extended at a large deviation from the polarity axis (e.g. 130 degrees), the localization of Rap1-GTP and the polarity axis changes by only 20 degrees and the cell makes just a small turn. Therefore, the stable localization of Rap1-GTP in a broad front allows for a relatively long maintenance of the polarity axis. In this respect the localization of Ras-GTP fluctuates much stronger as it is only present in the ultimate tip of the extending pseudopod and therefore its direction changes with each pseudopod. Wild-type cells occasionally extend a de novo pseudopod in the rear. Before such a pseudopod starts, many events are observed to occur at the place of future pseudopod extension that begin with an increase of Ras-GTP levels, followed by an increase of Rap1-GTP and the depletion of myosin filaments. These Ras-GTP-induced events lead to the local instability of the contractile cortex and simultaneously activates Rac/SCAR/Arp2/3 leading to polymerization of branched F-actin and the induction of a pseudopod. The probability that such an event occurs increases when Ras or Rap1 are more activated (as in the constitutive active Rap1G12V), when myosin is absent or depolymerizes more easily (as in cGMP mutants), or when the contractile cortex is less stable (such as in *for-AEH*-null cells).

When a wild-type cell extends a *de novo* pseudopod in the rear and this pseudopod grows, the cell gets two fronts with two polarity axes. We determined that the strength of the Old axis in the absence of pseudopod activity declines with a half-time of 88 seconds with 95% confidence interval of 63 to 148 seconds. This half-time of "forgetting" long-term memory is within the range of the measured dissociation half-time of the cGMP-GbpC complex in vivo (115 seconds; 95% confidence interval 93 to 149 seconds [40]), suggesting that long-term memory may

reside in the cGMP-GbpC complex. The half-time of information storage in this memory ("learning") is 25 ± 4.6 seconds (Fig 7), about two pseudopods.

The kinetic properties of long-term memory becomes interesting during cell aggregation where waves of cAMP attract cells. These waves increase during about 90 second and occur about every 5 minutes [41]. When a wave of cAMP hits the cell, the sudden strong increase of the cAMP concentration induces a cringe of the cell that is associated with a strong increase of cGMP levels and the depolymerization of myosin filaments in the entire cell [20, 42, 43]. This cringe is followed by the extension of a pseudopod in the direction of the cAMP gradient [42]. The Old polarity axis is annulated by the cGMP-mediated depolymerization of myosin filaments and a New polarity axis is induced by the new pseudopod. This suggest that the cringe erases the memory of the past ~10 pseudopod of the Old axis, whereas the fast 25 s half-time of memory storage by the New pseudopod allows to rapidly establish a New polarity axis in the direction of the cAMP gradient, that is present during about 90 seconds. When the cAMP wave has passed by, the memory of the new axis allows the cells to continue movement in the same direction, until they are hit by a new cAMP wave [44].

In summary, memory of the polarity axis is a Long-term memory that integrates total pseudopod activity in the past ~2 minutes using a cGMP/Rap1/myosin pathway. Memory leads to a polarized cell with suppressed pseudopods in the rear, leaving pseudopods to be extended in a global broad area of the front.

## Short-term memory of pseudopod position

This memory stores the position where a pseudopod is extended, and enhances the probability to start a new pseudopod at that position. To understand how this memory operates, its properties have to be combined with the characteristics of the pseudopod cycle [3, 15, 23]. When a pseudopod is induced, it grows at a constant speed and then stops. It is not exactly known why a pseudopod stops, but it could be related to build up of membrane tension in the front, local depletion of resources or the formation of a local inhibitor [15]. Whatever the mechanism, when a pseudopod stops, a cell rarely extends a new pseudopod at the position where the previous pseudopod stops, probably because the mechanism/inhibitor that led to the stop of that pseudopod is still present [3]. Thus, when pseudopod P1 stops, pseudopod P2 is extended elsewhere. When pseudopod P2 stops, it appears that this inhibition of pseudopod P1 has faded away and P1 has actually left behind a memory so that P3 is often extended from the position of pseudopod P1. And consequently, pseudopods P4 is extended from position P2, etc, leading to the alternating left/right extension of pseudopods. In unpolarized cells pseudopods P1 and P2 can be far apart, as much as 180 degrees, and unpolarized cells wiggle with little persistence of direction. In contrast, in polarized cells all pseudopods are formed in the front 30% of the cell, and therefore pseudopods P1 and P2 are nearby at a relatively small average angle of 55 degrees, and polarized cells move in a zig-zag fashion with strong persistence of direction.

When pseudopod P1 is extended for a shorter period, the probability decreases that pseudopod P3 is extended from the same position as pseudopod P1, suggesting that learning position requires time. The data show simple kinetics with a half-time of 3.2 ± 0.6 seconds. Since the average growth time of a pseudopod is 12 ± 6 seconds [3], learning is sufficiently fast to obtain accurate positional memory for >90% of the pseudopodia. The kinetics of forgetting is more complex. On average, pseudopod P3 starts 18 seconds after pseudopod P1 has stopped. We measured positional memory as function of this time interval, and observed that positional memory of pseudopod P1 is fully retained up to 28 ± 3 seconds after pseudopod P1 has stopped, and then positional memory gradually declines. At 28 seconds after the stop of pseudopod P1 about 90% of pseudopod P3 have already started (289 of the 322 pseudopods

analyzed) and thus have fully exploited the positional memory of pseudopod P1. At times longer than 28 seconds, memory is lost with a half time of 13 ± 2 seconds. At the moment that pseudopod P5 will start (on average at 48 seconds after pseudopod P1 has stopped) only 7.5% of the memory of pseudopod P1 is still present. Thus pseudopod P5 is largely directed by the positional memory of pseudopod P3 and only slightly by the positional memory of pseudopod P1. This analysis indicates that positional memory is short-term and remembers the position of only one pseudopod.

The positional accuracy of the memory can be estimated from the variance of the angle between pseudopod P1 and pseudopod P3, which is about 40 degrees and equivalent to about 11% of the circumference of the cell. Thus memory is about 90% accurate for direction. The underlying memory of position may be even more accurate, because previously [25] we have determined that the variance in pseudopod direction is composed from the variance of position where the pseudopod starts (~40%), the variance of local curvature of the membrane at that position (~40%%) and variance in extending pseudopod perpendicular to this curvature (~20%). Therefore, we suspect that the actual accuracy of the position memory may be > 90%. Importantly, cells can use positional information for directional movement, only because polarized cells have a very regular ellipsoid shape so that pseudopods that start at a specific position and are extended perpendicular to the surface will have predictable direction. Interestingly, we have shown that *dia*-null cells, which have a very irregular shape, exhibit splitting pseudopods with normal positional memory, but the local curvature of the membrane at that position is so irregular that pseudopods perpendicular to this curvature are extended in very different directions [25].

The molecular mechanism of this positional memory has long been elusive. We and others [11, 13] envisioned that at the place of the extending pseudopod a molecular marker is formed that remains present for some time after the pseudopod stops and that this marker enhances the probability of a new pseudopod at that position. Unpolarized cells lacking the long-term memory have normal short-term memory (Table 1), excluding the cGMP/myosin signaling pathway for positional memory. In addition, all ~100 signaling mutants investigated with defects in the Ras, Rap, Gα, PI3K, TOR, and myosin pathways have normal LR-bias. Recently, we obtained evidence that the LR-bias of pseudopod extension is associated with the alternating left/right activation of branched F-actin [13]. Activation of Ras and F-actin form a positive feedback loop. Spontaneous patches of activated Ras can induce F-actin filled protrusions, and *vice versa* spontaneous protrusions can induce local Ras activation. In alternating LR splitting pseudopods the formation of F-actin precedes Ras activation. Interestingly, cells treated with F-actin inhibitor LatA still possess spontaneous activated Ras patches, but they are no longer formed alternatingly to the right and left [13], suggesting that the molecular mechanism for positional memory occurs downstream of Ras and is closely associated with the machinery that induced branched F-actin in the extending pseudopod. Local activation of Arp2/3 and SCAR/WAVE play a prominent role in inducing branched F-actin and the start of a pseudopod [45]. Among several mutants investigated (Arpin, SCAR/WAVE, Pir121, Nap1) we identified the first mutant that lacks the alternating RL extension of splitting pseudopods: the phosphomimetic SCAR-SD expressed in *scar*-null cells [13]. Here it is shown that these cells are polarized with long-term memory and they frequently extend splitting pseudopods at the front, but these splittings are not alternatingly to the left and right. Detailed analysis revealed that they do not retain positional information. Thus pseudopod P3 is not made at the position of pseudopod P1, but at a much more random place in the front 30% of the cell. SCAR/WAVE is phosphorylated at multiple serines in the acidic C-terminal region of the protein; in mutant SCAR-SD these five serines in the C-terminal region are mutated to the negatively charged aspartate [23]. Detailed information on the local phosphorylation of SCAR/WAVE during and

after pseudopod extension is not available, but the biochemistry of SCAR/WAVE activation may provide a mechanism for this memory. Activation of the SCAR/WAVE complex by Rac involves at least two steps. The SCAR/WAVE complex is held in an auto-inhibited state by the acidic loop binding to a helix with positive charges [23, 45]. In unstimulated cells the acidic loop is phosphorylated thereby increasing the negative charges and keeping SCAR/WAVE locked in the inactivated state, which impairs activation by Rac. Dephosphorylation of the acidic loop loosens the SCAR/WAVE structure, allowing activation by Rac and induction of a pseudopod. After the pseudopod stops and SCAR/WAVE is no longer active it may finally fall back to the phosphorylated fully closed state via the intermediate unphosphorylated state; this intermediate unphosphorylated state is partly open and inactive, but easily activatable. We envision that at the position of an extinguishing pseudopod SCAR/WAVE remain present for some time in this partly open non-phosphorylated state, which forms the memory for the pre-ferred side for new activation. In the phosphomimetic SCAR-SD this local memory is disturbed because the intermediate state is absent. Although this model can explain the observed experiments, alternative models are possible in which SCAR-SD interferes with the memory encapsulated in the complex local regulation of branched F-actin by multiprotein complexes of Arp2/3 and nucleation promoting factors such as SCAR/WAVE, WASP and WASH [46].

In summary, short-term positional memory rapidly detects the local activity of one pseudopod, leading to the local accumulation of an enhancer of new pseudopod activity, possibly dephosphorylated SCAR/WAVE. This local enhancer triggers rather precisely the formation of the next-next pseudopod at that position.

## Coordination between memory of position and memory of polarity axis

For efficient persistent cell movement good coordination of pseudopod extension in time, space and direction is required. For that, coordination between memory of position and memory of polarity axis is essential. The refined temporal regulation of pseudopod extension ensures that cells extend only one pseudopod at a time [15]. Mutants defective in the contractile cortex extend multiple pseudopods in a temporal and spatial chaotic manner leading to very low persistence [26]. The very regular extension of one pseudopod allows cell to incorporate mechanisms that preserves directional extension. The two components that contribute to persistence -memory of position and memory of polarity axis- are coordinated and mutually reinforcing. Memory of position leads to two series of pseudopods that start at different positions, P1,3,5. . . and P2,4,6. . ., that leads to strong persistence only when they start nearby, which requires a polarized cell. Memory of polarity is not very precise, pseudopods start somewhere in the 30% front of the cell, and precision requires memory of position.

Memory of the polarity axis has a half-time of "forgetting" of 88 ± 43 seconds. Cells extend on average one pseudopod every 15 second. The 88 second half-time of memory implies that during this 15 seconds only ~11% of the memory of previous pseudopods is lost. Therefore, cells memorize 89% polarity of the previous pseudopod and 77% of the polarity of the pre-previous pseudopod, etc. The total memory is the sum $\sum_{n=0}^{n=\infty} 0.89^n = \frac{1}{1-0.89} = 9.0$. Thus the memory of the polarity axis is a long-term memory integrating information on the previous ~9 pseudopods, which are extended during the past ~2 minutes. Mutant SCAR-SD has only memory of polarity and exhibits a persistence of about 8.5 steps (Table 1).

Memory of position has a short half-life (Fig 4D and 4E): about 90% of pseudopod P5 remember the position of P3, but only 7.5% remember the position of P1. Although positional memory transmits positional information only to the next pseudopod (i.e. P1 to P3), indirectly the information is transmitted further from P3 to P5 and from P5 to P7, etc. How far does the information on the position of P1 penetrate to further pseudopods? As mentioned above, 90%

of pseudopods P1 transmit positional information directly from P1 to P3 with a positional accuracy of 90%. Thus, the information transfer from P1 to P3 equals $0.9^*0.9 = 0.81$, and the transmission from P1 to P5 equals $(0.81)^2$, and from P1 to P7, equals $(0.81)^3$, etc. The total indirect memory is the sum $\sum_{n=0}^{n=\infty} 0.81^n = \frac{1}{1-0.81} = 5.3$ pseudopods. Several mutants have only memory of polarity and exhibits a persistence of about 7 steps (Table 1).

Thus, the total direct long-term memory of the polarity axis (~9 pseudopods) and the indirect short term memory of position (~5 pseudopods) are well coordinated, and together they are responsible for the observed persistence of 15 steps. Interestingly, the indicators for memory of position (%LR) and polarity (%F) have similar values in four cell lines, the protist *Dictyostelium*, human neutrophils and mesenchymal stem cells and the fungus *B.d. chytrid*. These cells all have a persistence of about 13 to 16 pseudopods.

## Conclusions

Persistence of cell movement has been observed in many organisms. The present study suggests that persistence of cell movement is mediated by two very different types of directional memory: polarity and position. Polarity restricts new pseudopods to the front region of the cell and is mediated by a long-term memory of global pseudopod activity, while the alternating right/left extension of new pseudopods is mediated by a short-term memory of accurate positional information. The analysis of mutant in which either short- or long-term memory is lost reveal that the combination of short- and long-term memory make the cells exquisitely competent in persistent movement for optimal foraging and chemotaxis. The analysis tools for investigating memory in *Dictyostelium* were applied to human neutrophils and mesenchymal stem cells and the fungus *B.d. chytrid*, all showing similar characteristics of polarity and alternating right/left extension of pseudopods from the front of the cell. The synergy of short- and long-term memory explains their fundamental role in persistent movement for enhanced cell dispersal, food seeking and chemotaxis. *Dictyostelium* mutants begin to reveal the different molecular mechanisms of these directional memories and may guide the characterization of directional memories in other organisms.

## Materials and methods

### Cell lines and preparation

The cell lines used are the wild-type AX3, and the mutants *scar*-null cells with a deletion of the *scrA* DDB_G0285253 gene [23], SCAR-SD in *scar*-null [23], *gc*-null with deletions of the *gcA* DDB_G0275009 gene encoding GCA and of the *sgcA* DDB_G0276269 gene encoding sGC [38], *gbpC*-null with a deletion of the *gbpC* DDB_G0291079 gene [17], *myoII*-null with a deletion of the *mhcA* DDB_G0286355 gene [47], *forAEH*-null with deletions of the *forA* DDB_G0279607 *forE* DDB_G0269626 gene and the *forH* DDB_G0285589 gene [26] and *racE*-null cell with a deletion of the *racE* DDB_G0280975 gene [26]. RapAG12V cells are AX3 wild-type that express the dominant active RapAG12V from an inducible promotor, and express the sensors Ral-GDS-GFP and cytosolic-RFP [48]. The cell lines sGC-GFP and ΔNsGC-GFP express these proteins in *gc*-null cell [38]. Please note that previously [22, 49] we labeled mutant SCAR-SD wrongly as SCAR-S55D, which was due to a syntax error in our list of genetically modified organisms.

Cells were grown in HL5-C medium including glucose (ForMedium), containing the appropriate antibiotics for selection. Cells were collected and starved for 2–3 hours (vegetative cells) or 5–6 hours (starved cells). Cells were then harvested, suspended in 10 mM $KH_2PO_4$/$Na_2HPO_4$, pH 6.5 (PB), and used in experiments with microscopy recordings, using either

phase contrast (Olympus Type CK40 with 20x objective and JVC CCD camera) or confocal laser scanning microscopy (Zeiss LSM800; 63x numerical aperture 1.4 objective). Unless mentioned otherwise, the images were recorded at a rate of 1 frame per second and an x,y-resolution of 245 nm for phase contrast and 198 nm for confocal microscopy, respectively. The source of the movies of *Dictyostelium* wild-type and mutants, neutrophils, mesenchymal stem cells and *B.d. chytrid* is as described in [15] and is given in S2 Table.

## Pseudopod identification

The start and stop of pseudopod extension were identified as described previously [3, 15, 50], using the fully automated pseudopod tracking algorithm Quimp3, or a semi-automatic pseudopod tracking macro for ImageJ. In Quimp3 the active contour algorithm identified the contour of a cell in all frames as polygons of about 120 nodes. A pseudopod is defined as an outward convex deformation of a spherical cell, and was identified with an algorithm using minimal parameters for convexity, area extension and extension period. Then the algorithm goes back and forward in time to identify the frame number (f) in which extension starts and stops, respectively. Finally, the central node of the convex area in these two frames identifies the tip of the pseudopod at the beginning and end of the extension period, respectively. The program reports on position of the tip at start and stop. In semi-automatic pseudopod tracking, the investigator identifies the start and final position of a pseudopod growth. The custom made macro exports the position of the tip at start and stop, and prints a hard-copy arrow on the relevant frames of the movie. All cell lines were initially analyzed by both quimp3 and the semi-automatic method. High congruency of the two methods was obtained with the standard settings of quimp3 for all *Dictyostelium* mutants and Neutrophils, slightly modified setting for curvature and curvature gain for *B.d. chytrid*, while quimp3 was not effective with mesenchymal stem cells due to the very spiky appearance of the pseudopods.

To select cells for pseudopod analysis, first the displacement during 15 min of all 20–30 cells in the field of observation was determined, then the 3–5 cells were selected that have a displacement closest to the mean displacement, and finally the 2–4 cells were selected that remain attached to the substratum during the entire movie. About 30 pseudopods were recorded from one cell, about three cells were recorded from one movie and at least two movies were recorded for each strain or condition.

## Data analysis

For each pseudopod the position of the tip was identified in the frame immediately before extension started $(x_1,y_1,f_1)$ and again in the frame immediately after the extension terminated $(x_2,y_2,f_2)$. These data were used previously to calculate the primary pseudopod properties such as size, extension time, and direction of the pseudopod [15]. For the current study all pseudopods were annotated as *split* or *de novo*, and as left or right. A split pseudopod is defined as being formed at the side of a preceding pseudopod (its parental pseudopod), while a *de novo* pseudopod is formed in a region of the cell that was not previously part of a protrusion (see [50] for the algorithm used). Left (L) or right (R) is defined for splitting pseudopods that originate either at the left or the right side of the parental pseudopods, respectively. Using these annotations, the fraction of pseudopods that are formed in the front of a cell (%F) was defined as the fraction of splitting pseudopods of that cell. The sequence of a parental and two splitting pseudopods (de novo-split-split, or split-split-split) can be alternating LR or RL, or consecutive LL or RR. %LR is defined as the fraction of sequential splitting pseudopods that are alternating (LR+RL)/(LR+RL+LL+RR).

The temporal autocorrelation function of the number of extending pseudopods $ACF_\tau$ is the correlation of a time series with itself, shifted by time $\tau$. To determine the autocorrelation, the number of extending pseudopods during time $x_t$ was measured, yielding a mean $\bar{x}$ and a variance $\sigma^2 = <(x_t - \bar{x})>$. The autocorrelation function is given by $ACF_\tau = <(x_t - \bar{x})(x_{t+\tau} - \bar{x})>/\sigma^2$.

## Persistence

Persistence was determined with mean square displacement (MSD) from the position of pseudopods [3]. Because the method requires long trajectories for reliable estimates of the parameters [7, 51], the pseudopods of individual cells were concatenated to obtain a long synthetic sequence of pseudopods. The concatenation of the second to the first cell was as follows. The position and direction of all pseudopods of the second cell were moved in space and rotated such that the first pseudopod of this e second cell starts at the same position and is extended in the same direction as the last pseudopod of the first cell. This was repeated for all subsequent cells, yielding a trajectory of >250 steps/pseudopods.

The experimental data of pseudopod trajectories were well fitted by an exponential crossover from a correlated walk at short pseudopod intervals to a random walk at long intervals. The mean square displacement of the midpoint of pseudopod $i$ to pseudopod $i+n$ was determined and averaged over all pair of pseudopods,

$$MSD(n) = 2\lambda^2(nP - P^2(1 - e^{-n/P}))\tag{1}$$

where $\lambda$ is the characteristic step size in μm and $P$ is the persistence in number of steps. The experimental data with $n$ up to 80 steps were fitted to this equation using the least square method (lowest RSS) to obtain estimates for $\lambda$ and $P$ (S1 Table; Table 1).

## Statistics and reuse of data

Experimental data were analyzed using a linear model (e.g. Fig 4B), a non-linear model (e.g. Fig 9), or models with different number of parameters (e.g. Fig 8A). The parameter values of the model were varied to find the best fit between model and experimental data using the least residual sum of square (RSS) method. The goodness of the fit (95% confidence interval (95% CI)) for linear models was obtained from the linear regression. For non-linear models the goodness of the fit was estimated by bootstrap analysis with 30 random replacement of the data-set (some data points are not represented in the replicate and other data are represented twice) [52]; the 30 replicates were fitted yielding 30 estimates of the parameters with means and SD, where 95% CI = 1.96*SD. Statistical significance was tested with the Student's t-test.

For model discrimination the Akaike Information Criterion ($AIC_c$) and the F-test [53–56] were used to select the optimal model, i.e. the model with the lowest number of parameters that fit the data significantly better than models with the same or less parameters:

$AIC_c = 2p + N ln(RSS) + 2p(p+1)/(n-p-1)$ and $F = (N-p_2)/(p_2-p_1)*(RSS_2-RSS_1)/RSS_1$, where p is the number of parameters, N is the number of observations and the subscript 1 and 2 indicate the model with less and more parameters, respectively. The model with the lowest $AIC_c$ value is identified as the preferred model. The F-test was used to calculate the significance (P-value) of the difference between two models. The value of $exp((AIC_{c1} - AIC_{c2})/2)$ indicates how more probable the model 2 with more parameter is then model 1.

The original data on the dissociation of cGMP form GbpC in vivo (Fig 6 from [40]) was reanalyzed yielding and dissociation rate of 0.006 ± 0.001 s-1 (mean and 95% CI) yielding a half time of dissociation of 115 ± 28 s; interval 93 to 149 s.

## Theory and analysis of the polarity axis

Fig 5B reveals that the polarity axis is stabilized by pseudopod activity in its front. Therefore it is postulated that the a polarity axis $A$ is formed according to a conceptual scheme

$$1 - A + \Pi \underset{b}{\overset{a}{\rightleftharpoons}} A \tag{2}$$

where $\Pi$ is the pseudopod activity in the front that forms the axis; $a$ and $b$ are reaction constants of formation and inactivation of the polarity axis, respectively. Fig 5B suggests that pseudopod activity $\Pi$ is the product of pseudopod frequency and pseudopod growth rate with the units $\mu m/min^2$. To make $\Pi$ dimensionless, the actual measured pseudopod activity is normalized to the mean pseudopod activity [pseudopod frequency is $4.0 \pm 1.5$ pseudopods/min (mean and SD, n = 12 cells) and growth rate is $0.52 \pm 0.22$ $\mu m/s$ (mean and SD, n = 883 pseudopods; [15]) yielding mean pseudopod activity $\Pi_{mean} = 125$ $\mu m/min^2$]. Thus for each case analyzed $\Pi = \Pi_{observed} / \Pi_{mean}$.

The change of $A$ is given by

$$\frac{dA}{dt} = a\Pi(1 - A) - bA \tag{3}$$

When a cell extends a *de novo* pseudopod in the rear, the cell gets a new front with a New polarity axis, while the old front with its Old polarity axis is still present. Here the experimental cases are analyzed in which the New axis increases in strengths by pseudopod activity in its front, while the Old axis declines because it has no pseudopod activity. The New axis $A_n$ that appears at t = 0s and the Old axis $A_o$ are followed with time. The kinetics of formation of the New axis $A_n(t)$ and decline of the Old axis $A_o(t)$ are given by:

$$A_n(t) = A_n(\infty)[1 - e^{-(a\Pi_n + b)t}] \tag{4}$$

$$A_o(t) = A_o(0)e^{-bt} \tag{5}$$

Where $A_n(\infty)$ is the strength of the New axis at equilibrium, and $A_o(0)$ is the strength of the Old axis at t = 0. At time $T$ the increasing strength of the New axis equals the decreasing strength of the Old axis; we assume that at that moment the Old axis retracts, i.e. $T$ is the retraction time and $A_n(T) = A_o(T)$.

$$A_n(\infty)[1 - e^{-(a\Pi_n + b)T}] = A_o(0)e^{-bT} \tag{6}$$

$A_n(\infty) = a\Pi_n/(a\Pi_n + b)$. An estimate for $A_0(0)$ is obtained from the decline of the strength of the Old axis before a New axis is formed (Fig 5B) and the value for kinetic parameter $b$ obtained below, suggesting that the Old axis is about 90% of the strength of an axis with a mean pseudopod activity, i.e. $A_o(0) = 0.9a/(a+b)$. Then Eq (6) yields

$$\frac{1}{T} = (a\Pi_n + b)/\emptyset, \quad \text{where } \emptyset = -\ln\left(1 - \frac{0.9(a\Pi_n + b)}{\Pi_n(a + b)}e^{-bT}\right) \tag{7}$$

We analyzed 28 individual cases in which a cell starts a New axis with pseudopod activity $\Pi_n$, while the Old axis has no pseudopod activity and is retracted at time $T$. It appears that $\Pi_n$ and $T$ exhibit an inverse relationship, as shown by the linear plot of $1/T$ and $\Pi_n$ (Fig 7E). Although Eq (7) is non-linear, the inverse relationship suggests that the value of $\phi$ is nearly constant and does not strongly depend on $\Pi_n$ and $T$. The experimental data of $\Pi_n$ and $T$ were fitted to Eq (7) using the least square method (lowest RSS) to obtain estimates the kinetic

constants $a$ and $b$; the 95% CI was obtained by bootstrap analysis. Together this yields the optimal values and 95% CI: $a = 0.0268 \pm 0.0046$ s$^{-1}$ and $b = 0.0078 \pm 0.0032$ s$^{-1}$.

## Supporting information

**S1 Fig. Kinetics of positional memory, defined by %LR.**
(PDF)

**S2 Fig. The kinetics of turning in buffer and in a chemotactic gradient.**
(PDF)

**S3 Fig. Polarized cells without positional memory have a small alternating L/R bias.**
(PDF)

**S4 Fig. Persistence; parts 1–3.**
(PDF)

**S5 Fig. Path of four cell lines.**
(PDF)

**S1 Table. Persistence data.**
(PDF)

**S2 Table. Source of movies.**
(PDF)

## Acknowledgments

I am grateful to Ineke Keizer-Gunnink for the recording of many *Dictyostelium* mutants and to Arjan Kortholt for fruitful discussions.

## Author Contributions

**Conceptualization:** Peter J. M. van Haastert.

**Data curation:** Peter J. M. van Haastert.

**Formal analysis:** Peter J. M. van Haastert.

**Investigation:** Peter J. M. van Haastert.

**Methodology:** Peter J. M. van Haastert.

**Project administration:** Peter J. M. van Haastert.

**Resources:** Peter J. M. van Haastert.

**Software:** Peter J. M. van Haastert.

**Supervision:** Peter J. M. van Haastert.

**Validation:** Peter J. M. van Haastert.

**Visualization:** Peter J. M. van Haastert.

**Writing – original draft:** Peter J. M. van Haastert.

**Writing – review & editing:** Peter J. M. van Haastert.

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
