## [Decision Letter · Decision Letter 0]

28 Sep 2020

PONE-D-20-26226

Short- and long-term memory of moving amoeboid cells

PLOS ONE

Dear Peter,

Thank you for submitting your manuscript to PLOS ONE. Your manuscript has been reviewed by three expert referees, their comments and suggestions are attached. As you will realise, they differ slightly in their evaluation and criticisms, but they are unanimously recognising the value and the potential of a revised version to be acceptable. I am confident that you will find their expert criticisms and constructive suggestions helpful to revise your manuscript.

In summary, after careful consideration, we feel that it has merit but does not fully meet PLOS ONE’s publication criteria as it currently stands. Therefore, we invite you to submit a revised version of the manuscript that addresses the points raised during the review process.

We look forward to receiving your revised manuscript.

Kind regards,

Thierry Soldati, Dr Sci nat

Academic Editor

PLOS ONE

Journal Requirements:

"No. The funders had no role in study design, data collection and analysis, decision to publish, or preparation of the manuscript."

Reviewers' comments:

Reviewer's Responses to Questions

**Comments to the Author**

1. Is the manuscript technically sound, and do the data support the conclusions?

Reviewer #1: Partly

Reviewer #2: Yes

Reviewer #3: Yes

2. Has the statistical analysis been performed appropriately and rigorously? 

Reviewer #1: N/A

Reviewer #2: Yes

Reviewer #3: I Don't Know

3. Have the authors made all data underlying the findings in their manuscript fully available?

Reviewer #1: No

Reviewer #2: Yes

Reviewer #3: No

4. Is the manuscript presented in an intelligible fashion and written in standard English?

Reviewer #1: Yes

Reviewer #2: Yes

Reviewer #3: Yes

5. Review Comments to the Author

Reviewer #1: In this work, the author aims to describe the mechanism underlying persistent migration of dictyostelium cells. To do so, he relates the migration of the cells to the dynamics of pseudopods. He then aims at connecting the pseudopod dynamics to signaling pathways. I have two main problems with the manuscript:

1. Unfortunately, the author uses the language of memory to describe his findings. I found this utterly confusing, because it suggests some separate ‘module’ that stores the orientation of the polarization axis. In what he describes, however, this ‘memory’ is given by the elements that define the polarity axis; that is the pseudopods. This language made it very hard for me to differentiate between observations and (proposed) mechanisms. Also the level of description changes: sometimes the focus is on the molecular players, sometimes on the pseudopods and I could not decipher, where the mechanism lies. Eventually, I think, that ‘memory’ is just another word for ‘persistence’. I prefer the latter as it is much less prone to confusion.

2. The author has a tendency to base claims about mechanisms on correlations.

3. The model defined by Eq.(1) remained obscure to me: what is the inactive axis? What defines this axis? Why does it not appear in the equations? Suddenly, in the methods part on p. 25, the authors speaks about old and new axes that have not been (properly) introduced in this modeling section. How do they interact, if at all? How are they oriented? Their relative orientation is seemingly unimportant. Why?

Since many points remained unclear to me, I cannot recommend publication of this work in its current state.

Specific comments:

Abstract: “The direction of movement is not random, but is correlated with the direction of movement in the preceding minutes” There is no logical connection between the two parts of the sentence: randomness and correlation in the direction of movement are independent features. Notably, a movement can be random and still be correlated with the movement before. A classic example of such a behavior is a persistent random walk, described by Fürth 100 years ago. The author knows this as indicated by the first paragraph of the introduction, where he writes about “correlated random walks”. But then he contradicts hjmself by writing “persistence versus random movement”. Again, these two notions are not mutually exclusive.

P. 3, 2nd paragraph: “The timing and direction of pseudopod formation in amoeboid cells has been described as an ordered stochastic process” This process has also been described as resulting from deterministic dynamics: Dreher et al NJP 2014, Stankevicins et al, PNAS 2020

P. 3, 2nd paragraph: “this zig-zag trajectory provides persistence” Persistence is formally defined via the direction autocorrelation, which typically decays exponentially. The characteristic decay time is the persistence time. For an “ideal” zig-zag trajectory, the direction autocorrelation would be a periodic function. So, I am not sure what the author wants to say.

P/4, 1st paragraph: here the author introduces the notion of memory. It was not clear to me, whether memory was used as a synonym for persistence or whether this was a new concept. Since memory is used in different contexts with different meanings, I invite the author to clarify. If it is the former, I would suggest to not mention the word memory at all.

P.5, 1st paragraph: “the typical properties of a correlated random walk with a correlation/persistence of

about 11 pseudopods, representing the memory of this cell (Fig. 1b). The sequence of three pseudopods

is frequently alternating to the left and right, contributing to a persistent zig-zag trajectory” Just to repeat my previous comments: the same words are used in different contexts, which is confusing. Please, define clearly what you mean and then stick to your defined terms. - The same problem is visible throughout the manuscript and should be rectified.

P.6, last paragraph: “the periodic autocorrelation time is about twice the pseudopod interval. This means that the start of the first pseudopod is correlated with the start of the third pseudopod, while the start of the second pseudopod is correlated with the fourth pseudopod.” Agreed, but this does not discriminate between the two mechanisms, that is, whether the direction of P3 is determined by the position of P1 or P2. In fact there is a strong anti-correlation at about 15 s, which suggests that P2 is determined by P1 (and P3 by P2), if I follow the argument of the author.

p. 7: “The kinetics of learning and forgetting the position of splitting pseudopods” I really do not understand, why I should invoke the word learning in this context. If the cell just deposited a marker at the place where a pseudopod is formed, would you call this learning?

p. 9, 2nd paragraph: “pseudopod activity in the front is the input signal to generate, memorize and maintain a polarity axis.” From a correlation you infer a mechanism, but you do not have any evidence for a causal relation between pseudopod activity and persistence.

P. 10, 1st paragraph: “It should be noted that cGMP diffuses rapidly in the cell [28], and that cGMP induces an increase of myosin filament formation in the entire cell [17]. However, cell polarity is associated with increased

myosin filaments in the rear and reduced myosin filaments in the front of the cell [29].” How are these two sentence are meant to be understood? As an introduction to the rest of the paragraph; as pointing out a contradiction?

p. 11, 2nd paragraph: “Produced cGMP rapidly diffuses in the cell, activates GbpC leading to an increase of myosin filament in the entire cell. The second pathway also starts with Ras-GTP in the front and mediates the activation of Rap1 in the front half of the cell.” If I understood correctly, you have tracked the distributions of various proteins, but not their activities. How can you thus draw these conclusions about the spatiotemporal activation profiles?

p. 11, last paragraph: The way the model is referred to seems to me too vagua. The model should at least roughly be explained in the main text: what are the main assumptions, how are the assumptions implemented, how is the model analyzed?

p. 12, 1st paragraph: “For determining exact kinetics, only those 28 cases were used in which the Old front does not exhibit any pseudopod activity after the New front was made.” Why is it appropriate to only consider the described subset? Does the model with the thus determined parameter values also cover the kinetics of the other pseudopods?

Reviewer #2: The author describes a short- and a long-term memory in amoeboid cells. The short-term memory stores the information where the last pseudopod was formed for circa 20 seconds and is mediated by branched F-actin and the SCAR/WAVE complex. The long-term (2 min) memory depends on a cGMP-binding protein that established the cell rear by inducing the formation of myosin II filaments. This inhibits the pseudopod formation at the rear and promotes the formation at the front.

Major concerns:

I find it difficult to follow up the neutrophil, mesenchymal and B.d. chytrid data, because the paper is not accessible for me at the moment. It comes in the last paragraph a little bit out of the blue. The author could explain it a bit more why he has chosen those. It seems that neutrophils and Dicty are fast, while Bd and mesenchymal stem cells are slow. It is PLOS so he can write as much as he wants.

Scar S55D phospho-mutant – Why does in particular this mutant does not split pseudopods (alternating left and right) like wild-type cells? I found their JCS paper, but that also does not tell me more. It would make reading easier with a bit of background. They refer to a Current Biology paper, but the description of the mutants does not agree, neither is there anything on dictybase. Or is it S5D from Ura et al? Since it is the only mutant he looked at, that has problems with memory positioning it would be important to know. It appears to be in the SHD domain.

It is essentially not new data, but I like the quantification he has done and how he combined the long and short memory story to explain directed movement. So good enough for this type of submission. To measure the angles is a good way to access forward movement using pseudopod splitting.

Data analysis is done carefully and the timely separation of the activation of ras, rap1 etc. is quite impressive. The data quality looks good and is convincing for me. Like the turning experiments, too. The finishing model is also a good summary.

Minor concerns:

Maybe call it SCAR/WAVE, and reference accordingly, then maybe also people outside the Dicty community look at the paper?

The switching between left and right, North and East to describe pseudopod formation is a bit confusing. It is sometimes difficult to follow if developed or vegetative cells have been used for the analysis.

Page 3:

bF-actin – branched F-actin

cGMP binding protein – name it here directly gbpC?

… extend pseudopods in the rear 70% of the cell, because ... - … extend pseudopods in the rear (70% of the cell), because... (make it easier to read)

Page 4:

Scar/Factin – SCAR/WAVE/F-actin

Table 1:

Are forAEH, racE and rap1G12V cells starved or vegetative? What is the purpose of the blue arrow in the last column connecting scar S55D vegetative and starved?

Figure 7:

The capitalisation of Old and New looks a bit odd and unnecessary.

Reviewer #3: This is a nice work demonstrating the ordered pattern of successive pseudopod extension demonstrating a positional memory (pseudopod P3 remembering the position of pseudopod P1 and anticorrelated with pseudopod P2…) and how it is related to the polarity axis memory and to the regulation of the acto-myosin machinery.

The experiments suggest that a lower pseudopod activity in the front is the input signal to generate a new (de novo) pseudopod at a random new location, and hence modifies the polarity axis. The mechanism proposed in the discussion (Fig. 9) with the role of the signaling pathways with Ras-GTP and Rap-GTP in the regulation of this polarity are convincing as well.

The paper is well written in particular the introduction.

One problem with this paper is the fact that the majority of pseudopod data are identical to those used in ref [15 = van Haastert, P.J.M. (2020). Unified control of amoeboid pseudopod extension in multiple organisms by branched F-actin in the front and parallel F-actin/myosin in the cortex. Submitted].

This reference is not deposited on a reprint server so it is difficult for me to evaluate the novelty of the current work, and especially to evaluate raw data for the biophysical part I can better judge.

For instance, I would like to watch movies but none is included here.

Mean squared displacements (MSD) fits are used to fill the third column of Table 1 (persistence time) and these important data are discussed at several occasions.

The origin of the MSD data (except a short example given in Fig. 1B) is not discussed.

Is it from the same submitted ref [15]?

The author should present all raw MSD curves (11 conditions=11 panels) with fits in supplementary information to evaluate their quality.

Usually reliable MSD analysis need a lot of statistics: either very long trajectories (up to 20h) as in Li and Cox (PLoS ONE 2008) or very numerous cell tracks (several hundred lasting about 1 h) as in Gole et al. (PLoS ONE 2011). I am surprised that sometimes there is only 8 cells analysed at 1 frame/sec making difficult to record long enough trajectories to analyze rigorously their persistence time.

In the same line of argument, how is calculated the error on the fitted persistence time in Table 1 ?

Page 19, I am not convinced by the conclusion that pseudopod P3 is slightly biased toward P2 when P2 direction is far from P1 direction. This conclusion holds from a better non linear fit with two fitting parameters than a linear one with one fitting parameter. But it is a general rule that the more fitting parameters, the better the fit !

Looking at the vertical scatter of raw data for a given ph1,2, the ratio between the black dot scatter width and the green bias width is at least 2 ! One possibility would be to estimate the error bar on the two parameters of the non linear fits to learn how reliable is this parameter. Perhaps there is a more systematic procedures to evaluate the significance of the proposed fits.

Minor comments.

A missing dot page 6 after [6,7,11]

Explanation of the first order kinetics is unclear (data of Figs. 4b,e, introduced in page 13). Why x-scale in 4B goes only to 12s, not 25s as in 4a ?

Page 4: Don't we expect a lower %LR for vegetative than polarized cells ?? Please comment.

The pseudopods intervals (in s, or min) for neutrophils, mesenchymal stem cells and B.D. chytrid should be indicated in Table 1.

I assume these cells have different intervals but the same persistence time in pseudopods number. I feel this can be better discussed in the paper.

On the model page 18: I don't understand the sentence "Simulations reveal that phi depends predominantly on the kinetic parameters a and b". -> Which simulations ? -> What about calculating and plotting, phi as a function of Pn, once parameters a and b are calculated with

the current hypothesis that phi is independent of Pn and T ? This is a kind of self consistent-argument.

6. PLOS authors have the option to publish the peer review history of their article (what does this mean?). If published, this will include your full peer review and any attached files.

Reviewer #1: No

Reviewer #2: No

Reviewer #3: No

---

## [Decision Letter · Decision Letter 1]

23 Dec 2020

PONE-D-20-26226R1

Short- and long-term memory of moving amoeboid cells

PLOS ONE

Dear Peter,

Thank you for submitting your manuscript to PLOS ONE. The three referees and myself have the clear impression that we are making progress ... but we differ in our fine evaluation. One referee is fully satisfied with the revisions and one is satisfied that the PLOS criteria are met, but nonetheless has "regrets" about some of your decisions. The third referee is less enthusiastic and clearly details, in a constructive end expert manner the points that still need your attention before the manuscript canoe accepted for publication.

We look forward to receiving your revised manuscript.

Kind regards,

Thierry Soldati, Dr Sci nat

Academic Editor

PLOS ONE

Reviewers' comments:

Reviewer's Responses to Questions

**Comments to the Author**

1. If the authors have adequately addressed your comments raised in a previous round of review and you feel that this manuscript is now acceptable for publication, you may indicate that here to bypass the “Comments to the Author” section, enter your conflict of interest statement in the “Confidential to Editor” section, and submit your "Accept" recommendation.

Reviewer #1: All comments have been addressed

Reviewer #2: All comments have been addressed

Reviewer #3: (No Response)

2. Is the manuscript technically sound, and do the data support the conclusions?

Reviewer #1: Yes

Reviewer #2: Yes

Reviewer #3: Partly

3. Has the statistical analysis been performed appropriately and rigorously? 

Reviewer #1: Yes

Reviewer #2: Yes

Reviewer #3: I Don't Know

4. Have the authors made all data underlying the findings in their manuscript fully available?

Reviewer #1: Yes

Reviewer #2: (No Response)

Reviewer #3: No

5. Is the manuscript presented in an intelligible fashion and written in standard English?

Reviewer #1: Yes

Reviewer #2: (No Response)

Reviewer #3: Yes

6. Review Comments to the Author

Reviewer #1: I thank the author for his responses. I do not think that the definition of memory and persistence agrees with the ones used in the community I am mostly frequenting, but since he defines all terms in a box, I can live with it. Still I do not think that it is a good idea to deviate from the common terminology - even if one has good reasons to not like it.

The discussion between randomness and deterministic dynamics requires probably an in-depth discussion. Here, I would just like to mention that it is impossible to start with exactly the same initial condition twice and that hence the outcomes can be VERY different if the system is chaotic. In any case "ordered stochastic process" does not seem to be a well-defined concept.

I fear that the terminology used by the author will rather generate confusion than help to understand amoeboid motility. Alas, this is what the author chose to do. According to my understanding of the publication criteria of PLoS one, this is not a reason not to recommend publication. So, since all technical issues have been addressed by the author, I do recommend to accept the manuscript for publication.

Reviewer #2: All corrected now.

I am kind of glad, that they had mislabelled Ura's mutant and it wasn’t me being unable to find it...

Reviewer #3: The author has changed significantly the paper taking into account many of the reviewer’s comments including mine. It is much more clear now.

However, I feel some important points should still be considered and I cannot recommend this work in its present state.

1) It is very valuable to have access to all raw MSD (new Fig. S4). Generally the quality of the data set seems reasonably good to some exceptions: (i) the solid and dashed lines are very far at both short and long time steps for the starved gbpc-null data (part 2). (ii) The mesenchymal and the B.d. Chytrid data are not very well fitted by the Furth model at any time scale as well (Eq. 1): the data alternate above and below the Furth model and I am surprised by the low error on the fitted parameter. For instance, if we fit the data between n=0 and n=50 instead of 75, I am sure the asymptotic P value will be lower than the range indicated in Table S1. The author uses a bootstrap analysis (without detailing it) due to the non linear nature of the model. I will be curious to compare the resulting error on P with another method using a linear asymptotic fit (dashed line, so fitting between for instance n=30 and 50/60/70/80 for a comparison) and a 95% confidence interval method.

2) The Eq (1) is a Furth formula written is an unconventional way with lambda the step length. Usually, one uses the diffusion constant D or the speed V or the persistence length d= P lambda = V tp where tp is the persistence time (in s or in min, actually refered as “P, time” in min in Table S1). I recommend a more conventional form in addition to the one given by the author with his step unit (Eq 1).

The notation < ,+ ^2 > has to be changed as the index “i” is an averaged variable, and the only remaining variable is n which is the time t in steps, n=t/int (where "int" is the interval reported in Table S1). Do not use capital D which is usually reserved for the diffusion constant (which is important to be reported as well, see my next comment). I suggest for instance r^2(n) or MSD(n).

(^ means exponent)

2) The diffusion constant D=(V^2 tp)/2=(P lambda^2)/(2 int) for each cell line should be reported in Table S1 in µm^2/min.

By looking at the data, I am calculating D=275 µm^2/min, 81 µm^2/min and 63 µm^2/min for starved Dd, vegetatitive Dd and neutrophils respectively. I am a bit surprised by the low value for the neutrophils which in my mind was even faster than Dd. The author should compare their value with literature data.

3) For the other organisms investigated, the only reference to the source of the movies in the new Table S2 is a bit short for this reviewer. I cannot evaluate without seeing the raw data (and these raw data are not present as well in the companion PLoS ONE paper recently published by the author) if the same concept of polarity and memory, with rear, front specificities is explaining convincingly their crawling motion.

It will be important at least in SI to show some trajectories with the alternating zig-zag front protrusions such as the one of Fig. 1a for these neutrophils, mesenchymal or Bd Chytrid cells. Finally, please show how the algorithm Quimp3 is detecting protrusions for these organisms.

4) Experimental conditions are missing with these other organisms: cell types (primary cells or cell lines like HL60...), recording conditions, composition of the culture medium...etc ? These information are not present as well in the other PLoS ONE paper.

4) I am not satisfied with the response to my comment:

“Page 19, I am not convinced by the conclusion that pseudopod P3 is slightly biased toward P2 when P2 direction is far from P1 direction. This conclusion holds from a better non linear fit with two fitting parameters than a linear one with one fitting parameter. But it is a general rule that the more fitting parameters, the better the fit !

Looking at the vertical scatter of raw data for a given ph1,2, the ratio between the black dot scatter width and the green bias width is at least 2 ! One possibility would be to estimate the error bar on the two parameters of the non linear fits to learn how reliable is this parameter. Perhaps there is a more systematic procedures to evaluate the significance of the proposed fits.”

The author answered:

“Using the Akaike Information Criterion (AIC) the polynomial model describes the experimental data 90-times better than the linear model (see legend Fig. 8a).”

I am not able to evaluate that answer but I really feel suspicious about this magical mathematics. Does a model with a third parameter fit 900-times better the data ? I suggest the editor to ask some specialist in statistics to evaluate this statement.

Minor comments.

The autocorrelation of Fig 3b is not defined: please define it and explain as well how is estimated the pseudopod interval as these two quantities are compared.

Fig 4: what is the difference between dark and gray bullets ?

Fig S4, part 1, there is two “Dictyostelium” graphs (I guess starved WT ) but next page in Fig S4/part 2 there are again "starved WT" data presented.

7. PLOS authors have the option to publish the peer review history of their article (what does this mean?). If published, this will include your full peer review and any attached files.

Reviewer #1: No

Reviewer #2: No

Reviewer #3: No

---

## [Editor Report · Decision Letter 2]

18 Jan 2021

Short- and long-term memory of moving amoeboid cells

PONE-D-20-26226R2

Dear Peter,

I am pleased to inform you that your manuscript has been judged scientifically suitable for publication and will be formally accepted for publication once it meets all outstanding technical requirements.

Kind regards,

Thierry Soldati, Dr Sci nat

Academic Editor

PLOS ONE
---

## [Editor Report · Acceptance letter]

27 Jan 2021

PONE-D-20-26226R2 

Short- and long-term memory of moving amoeboid cells 

Dear Dr. van Haastert:

I'm pleased to inform you that your manuscript has been deemed suitable for publication in PLOS ONE. Congratulations! Your manuscript is now with our production department. 

Kind regards, 

on behalf of

Dr. Thierry Soldati 

Academic Editor

PLOS ONE